# Palaeoenvironmental response of mid-latitudinal wetlands to Paleocene-Early Eocene climate change (Schöningen lignite deposits, Germany)

Katharina Methner[1], Olaf Lenz[2], Walter Riegel[2], Volker Wilde[2], Andreas Mulch[1,3]

[1]Senckenberg Biodiversity and Climate Research Centre, Frankfurt am Main, 60325, Germany
[2]Senckenberg Research Institute and Natural History Museum Frankfurt, Frankfurt am Main, 60325, Germany
[3]Institute of Geosciences, Goethe University Frankfurt, Frankfurt am Main, 60438, Germany

*Correspondence to*: Katharina Methner (katharina.methner@senckenberg.de)

**Abstract.**

The early Paleogene is marked by multiple negative carbon isotope excursions (CIEs) that reflect massive short-term carbon cycle perturbations that coincide with significant warming during a high-$pCO_2$ world, affecting both marine and terrestrial ecosystems. Records of such hyperthermals from the marine-terrestrial interface (e.g. estuarine swamps and mire deposits) are, therefore, of great interest as their present-day counterparts are highly vulnerable to future climate and sea level change. Here, we assess paleoenvironmental changes of mid-latitudinal late Paleocene-early Eocene peat mire records along the paleo-North Sea coast. We provide carbon isotope data of bulk organic matter ($\delta^{13}C_{TOC}$), organic carbon content (%TOC), and palynological data from an extensive peat mire deposited at a mid-latitudinal (ca. 41 °N) coastal site (Schöningen, Germany). The $\delta^{13}C_{TOC}$ data show a carbon isotope excursion of -1.3 ‰ (mean decrease in $\delta^{13}C_{TOC}$; -1.7 ‰ at the onset of CIE) coeval with a conspicuous *Apectodinium* acme. Due to the exceptionally large stratigraphic thickness of the CIE at Schöningen (10 m of section) we established a detailed palynological record that indicates only minor changes in paleovegetation leading into and during this event. Instead, paleovegetation changes mostly follow natural successions in response to changes along the marine-terrestrial interface. The available age constraints for the Schöningen Formation hamper a solid assignment of the detected CIE to a particular hyperthermal such as the Paleocene-Eocene Thermal Maximum (PETM/ETM1) or any succeeding hyperthermal event such as the Eocene Thermal Maximum 2 (ETM2).

Compared to other near-by peat mire records (Cobham, UK; Vasterival, F) it appears that wetland deposits around the Paleogene North Sea have a consistent CIE magnitude of ca. -1.3 ‰ in $\delta^{13}C_{TOC}$. Moreover, the Schöningen record shares major characteristics with the Cobham Lignite PETM record, including evidence for increased fire activity prior to the CIE, minor plant species change during the hyperthermal, a reduced CIE in $\delta^{13}C_{TOC}$, and drowning of the mire (marine ingressions) during much of the Schöningen CIE event. This suggests that either the Schöningen CIE reflects the PETM or that early Paleogene hyperthermals similarly affected paleoenvironmental conditions of a major segment of the paleo-North Sea coast.

# 1 Introduction

Early Cenozoic short-term hyperthermals such as the Paleocene–Eocene Thermal Maximum (PETM) or the Eocene Thermal Maximum 2 (ETM2) reflect rapid global warming events that punctuate the already warm Paleogene climate. These events coincide with perturbations in the global carbon cycle, manifested in negative carbon isotope excursions (CIE) due to a rapid and massive $^{13}$C-depleted carbon input into the ocean-atmosphere system. CIEs are a recurrent phenomenon of early and middle Eocene climate dynamics (e.g. Kennett and Scott, 1991; Cramer et al. 2003; Lourens et al. 2005; Zachos et al., 2003; 2008, 2010; Sluijs and Dickens, 2012).

Among the early Cenozoic climate perturbations, the Paleocene-Eocene Thermal Maximum (PETM) at ~ 56 Ma is one of the best-investigated Cenozoic hyperthermals with local warming of up to 10°C (e.g. Dunkley Jones et al., 2013; Frieling et al. 2014; Sluijs et al., 2006, 2011; Zachos et al., 2003). In general, the PETM-related CIE consists of a rapid onset, the main body of the excursion and a recovery composed of an early rapid phase followed by a more gradual increase in $\delta^{13}$C values (McInerney and Wing, 2011). Despite this general pattern, the magnitude of the CIE varies largely between different environmental settings (on average -2.5 to -5.5 ‰) and within individual records (-0.6 to -8.0 ‰) (see compilation of McInerney and Wing, 2011). The CIE is typically larger in continental than in marine depositional environments. Subsequent to the PETM, the ETM2 at ~ 54 Ma represents the next prominent hyperthermal similarly associated with rapid $^{13}$C-depleted carbon input to the ocean-atmosphere system resulting in a marine CIE of ~ -1.5 ‰ and global warming by several degrees (~ 3-5°C) (Cramer et al. 2003; Lourens et al. 2005; Sluijs et al. 2009; Stap et al. 2010; Frieling et al. 2014).

Multiple hypotheses exist to explain the $^{13}$C-depleted carbon input to the atmosphere-ocean system during the early Cenozoic, including methane clathrate destabilization on continental slopes (Dickens et al., 1995), thermogenic methane formation by magma injections into organic-rich mudstones (Svensen et al., 2004), oxidation of vast amounts of organic matter by drying of epicontinental seas (Higgins and Schrag, 2006), orbitally triggered permafrost thawing (DeConto et al., 2012), and wildfires and burning of peatlands (Kurtz et al., 2003) due to natural causes or triggered by a meteorite impact (Cramer and Kent, 2005; Kent et al., 2003). The magnitude of CIEs, if accounted for (1) variable preservation of the CIEs in terms of magnitude and duration in different archives (Trampush and Hajek, 2017; Lyons et al. 2019) and (2) carbon source effects resulting in local variations of CIEs (Sluijs and Dickens, 2012), potentially reveals the source of carbon input to the ocean-atmosphere system. Especially the mixing of terrestrial and marine organic carbon exerts a strong effect on the preserved CIE and calls for careful disentangling of the carbon isotope signal between these two components (Sluijs and Dickens, 2012).

Thus, assessing duration and magnitude of CIEs related to global carbon cycle perturbations at the marine-terrestrial interface is of great interest, as near coastal ecosystems are especially vulnerable to global climate change and sea level rise causing large ecological and economic threats (IPCC, 2014). Near-coastal wetlands play a major role in the global carbon cycle by storing large quantities of terrestrial organic carbon (e.g. Raghoebarsing et al., 2005; Rumpel et al., 2018), but simultaneously are a primary source of methane emissions to the atmosphere (Christensen et al., 2003; Kirschke et al.,

2013). Peatland conservation has thus become one of the pressing tasks to meet the Paris Agreement (Rumpel et al., 2018) as under global warming peatlands are likely to contribute significantly to future $CO_2$ emissions (e.g. Dorrepaal et al., 2009; Rumpel et al., 2018) and likely have done so during the PETM (Pancost et al., 2007). Extensive wetlands with peat forming mires and swamps were widespread in the European realm during the Cenozoic and may have significantly contributed to Cenozoic climate evolution on a global scale (e.g. Kurtz et al., 2003; Pancost et al., 2007).

Here, we explore carbon isotope ratios of bulk organic matter ($\delta^{13}C_{TOC}$) and the total organic matter content (%TOC) from the lower part of the Paleocene-Eocene Schöningen Formation (Germany). We evaluate the corresponding palynological record to assess the paleoenvironmental evolution of this wetland and compare our results to associated Paleogene lignite records along the paleo-North Sea (Cobham, UK; Vasterival, F) to identify potential regionally coherent paleoenvironmental patterns.

## 2. Study site and age constraints of the Schöningen Formation

The Schöningen opencast mine (Northern Germany) is the type locality of the Schöningen Formation (Riegel et al., 2012) and offers the opportunity to study Paleocene-Eocene climate change in wetland deposits along the paleo-North Sea coast (Fig. 1a) (Riegel et al., 2012). The Schöningen Formation is mainly early Eocene in age but probably includes the topmost part of the Paleocene. It comprises ~ 150 m of alternating lignite seams and clastic interbeds (Fig. 1b) (Brandes et al., 2012; Osman et al., 2013; Riegel et al., 2012). Detailed age constraints of the Schöningen Formation are weak and restricted to dinocyst assemblages and scattered radiometric ages from a core at Emmerstedt ca. 20 km N of Schöningen that were tentatively correlated to the Schöningen lignite succession (Riegel et al., 2012). It has been speculated that the lower part of the Schöningen Formation covers the PETM as the shallow marine deposits of Interbed 2 exhibit a conspicuous peak in the abundance of dinoflagellate cysts of the genus *Apectodinium* (Riegel et al., 2012). The occurrence of high abundances of the species *Apectodinium augustum* in mid- to high-latitudinal sediments (Schöningen paleolatitude is ~ 41°N (van Hinsbergen et al., 2015); see also Supplementary Information table S1) has been proven to be indicative of the PETM (Bujak and Brinkhuis, 1998; Crouch et al., 2001; Heilmann-Clausen et al., 1985; Iakovleva et al., 2001; Sluijs and Brinkhuis, 2009; Sluijs et al., 2006, 2007). The dinoflagellate cysts of the marine influenced Interbeds 1 and 2 only allow for an assignment of these deposits to dinocyst zone D5, which is characterized by the mass occurrence of *Apectodinium* spp. and includes the Paleocene/Eocene boundary (Ahrendt et al., 1995; Köthe, 2003). A further distinction in subzones D5na (uppermost Paleocene) and D5nb (lowermost Eocene) is based on the last occurrence of *Apectodinium augustum* (Köthe, 2003) (now *Axiodinium augustum*; Williams et al., 2015) which is the marker species for the PETM in open marine environments (Bujak and Brinkhuis, 1998). This species has not been found in samples from the interbeds at Schöningen and is also unknown from other Paleocene/Eocene records in Northern and Central Germany (Köthe, 2003). Therefore, unequivocal proof for the PETM as based on the occurrence of. *A. augustum* does not exist for the lower part of the Schöningen Formation. However, *A. augustum* may have not been present in marginal marine areas under reduced salinities and could therefore be absent from

the *Apectodinium* acme at Schöningen for pure ecological reasons. Frequent occurrences of *Apectodinium* species have also been noted at other times during the late Paleocene and early Eocene (Bijl et al., 2013, Frieling et al., 2018, Heilmann-Clausen, 2018). In the North Sea basin they existed in geographically restricted and marginal marine areas similar to the paleoenvironment at Schöningen, while at the same time *Apectodinium* was missing in nearby open marine settings (Heilmann-Clausen 2018). In the Schöningen record for example, another but smaller *Apectodinium* acme could be noticed below Seam 6 (Riegel et al., 2012). Increased nutrient supply and reduced salinity in the marginal marine environment may have favored *Apectodinium* which was therefore not necessarily associated only with warming events such as the PETM (Heilmann-Clausen, 2018).

## 3 Material and methods

### 3.1 Studied sections and sampling

Active lignite mining (1978-2016) yielded excellent exposures in the now-abandoned opencast mines at Schöningen (Fig. 1a, Supplementary Information S1 and Fig. S1). This allowed for dense sampling of the Paleocene-Eocene Schöningen Formation (Fig. 1b) in the western rim syncline of the NW-SE trending Helmstedt-Staßfurt salt wall. From the >4000 samples, collected from more than 50 individual sections over ~ 30 years accompanying the mining activities in Schöningen-Südfeld, we selected 121 samples for isotopic analyses. These samples comprise two of the lower lignite seams (Seam 1 and Seam 2) and the corresponding clastic interbed, presumably covering the latest Paleocene and early Eocene. The ca. 16 m record consists of three individual sections recovered laterally within 50 m. Stratigraphic continuation could be ensured by the well exposed, undisturbed and laterally traceable lignite seams (e.g. Riegel et al., 2012). The transitions between the seams and the marine interbeds are rather abrupt (cf. Supplementary Information S1 and Fig. S1), however, there is no evidence for any major hiatus. In order to get a more comprehensive picture of environmental and vegetation change in the latest Paleocene/early Eocene, samples from the underlying sediments (Main Seam and Interbed 1) are included in our palynological analyses.

The lower three seams (Main Seam, Seam 1 and 2, Fig. 1b) resemble each other in their petrographic and palynological characteristics. In general, the lignite seams are composed of an alternation of dark and medium brown layers, which often have tree stumps at their base and layers or lenses of charcoal with tissue preservation in the coal matrix (Riegel et al., 2012; Robson et al., 2015). Silts to medium grained sands dominate the clastic interbeds. There is still debate about prevailing deposition conditions of the clastic interbeds (Osman et al., 2013; Riegel et al., 2012). Interbed 1 includes *Apectodinium* spp. mostly in low numbers and shows little evidence for fully marine conditions but rather indicates local emergence (occurrence of driftwood and occasional rooting). In contrast, Interbed 2, containing rich dinocyst assemblages with peak abundances of *Apectodinium homomorphum* and other *Apectodinium* species (Fig. S4), is indicative of shallow marine depositional conditions (Riegel et al., 2012). For a more detailed description of the lithology of the sampled sections, the

reader is referred to Riegel and Wilde (2016) and Riegel et al. (2012) as well as to the detailed logs in the Supplementary Information (S1, Fig. S1).

## 3.2 Carbon isotope analyses

121 samples were selected for analysis of total organic carbon content (%TOC) and carbon isotope composition of bulk organic matter ($\delta^{13}C_{TOC}$), providing a %TOC and $\delta^{13}C_{TOC}$ record of 16 m with average sample spacing of ~ 13 cm. Sample preparation included freeze drying, grinding, removal of inorganic carbon (using 10 % HCl for 24 h at 40°C), centrifugation (4x at 2800 to 3000 rpm for 4 to 8 min) and sample drying (24 h at 40°C). About ~0.2 mg (lignite samples) and up to ~ 6 mg (marine interbed samples) were analyzed using a Flash EA 1112 (Thermo Finnigan) coupled to a MAT 253 gas source mass spectrometer (Thermo Finnigan) at the Goethe University - Senckenberg BiK-F Joint Stable Isotope Facility (Frankfurt). USGS 24 and IAEA-CH-7 standard materials were analyzed on a daily basis and replicate measurements of reference materials and samples indicate uncertainty of < 0.2 ‰ for measured $\delta^{13}C_{TOC}$ values. Total organic carbon concentrations [in %] were calculated by relating the signal size of the samples and the averaged signal size of the daily standards (USGS 24, n = 8). The typical error is ~ 0.5 % based on mass spectrometric analysis and the maximum difference in TOC contents of replicate measurements (including weighing uncertainties) was ~ 7 %.

## 3.3 Palynological analyses

The palynological analysis is primarily based on revised data of 59 samples from a section between the top of the Main Seam and the top of Seam 2 (Hammer-Schiemann, 1998). For palynological processing, lignite samples were crushed to a particle size of 1 to 2 mm. All lignite and interbed samples were carefully treated with hot 15 % hydrogen peroxide ($H_2O_2$) and ca. 5 % potassium hydroxide (KOH) for 1 to 2 h. To our experience this does not alter the palynomorph record to a major extent. The clastic samples were further treated with 30 % hydrofluoric acid (HF) for several days. HF was removed by 5 to 6 steps of decanting and diluting. All samples were sieved through a 10 μm-mesh sieve. Residues are stored in glycerine and permanent glycerine jelly slides were made. To obtain a representative dataset, at least 300 individual grains of pollen and spores were counted per sample at ×400 magnification (data in Supplementary Information table S3). The palynomorphs were mainly identified based on systematic-taxonomic studies of Thomson and Pflug (1953), Krutzsch (1970), Thiele-Pfeiffer (1988), Nickel (1996), Hammer-Schiemann (1998) and Lenz (2005). The simplified pollen diagram shows the abundance of the most important palynomorphs in percentages. Pollen and spores were calculated to 100 % whereas dinoflagellate cysts (*Apectodinium* spp.) and other algae were added as additional percentages (in percent of the total pollen sum).

To reveal compositional differences between the different coal seams and interbeds, non-metric multidimensional scaling (NMDS; Shepard, 1962a, b; Kruskal, 1964) was performed for the pollen and spore data set excluding algae. NMDS is the most robust unconstrained ordination method in ecology (Minchin, 1987) and has been successfully applied to palynological data in previous studies (e.g. Oswald et al., 2007; Jardine and Harrington, 2008; Mander et al., 2010; Ghilardi and O'Connell,

2013; Broothaerts et al., 2014; Lenz and Wilde, 2018). The method avoids the assumption of a linear response model or a unimodal response model between the palynomorph taxa and the underlying environmental gradients and avoids the requirement of normality of data. For NDMS the Bray-Curtis dissimilarity and the Wisconsin double standardized raw data values have been used (Bray and Curtis, 1957; Gauch and Scruggs, 1979; Oksanen, 2007). Wisconsin standardization scales the abundance of each taxon to its maximum value and represents the abundance of each taxon by its proportion in the sample (Mander et al., 2010). This equalizes the effects of rare and abundant taxa and removes the influence of sample size on the analysis (Von Tongeren, 1995; Jardine and Harrington, 2008).

## 4 Results and Discussion

### 4.1 The CIE in the basal Schöningen Formation

The total organic carbon content across the analyzed section ranges from ~ 0.2 % to 69 % and correlates with lithology generally with TOC contents >50 % in the lignite seams and <10 % in the clastic interbed (Fig. 2a). Carbon isotope ratios of bulk organic matter range from -25.1 ‰ to -28.3 ‰ (Fig. 2b). In the basal part (0 to 2.9 m) of the section $\delta^{13}C_{TOC}$ varies between -25.7 ‰ and -27.4 ‰ (average $\delta^{13}C_{TOC}$ = -26.76 ±0.46 ‰, n=26). At 3.0 m of section, $\delta^{13}C_{TOC}$ abruptly decreases to values as low as -28.3 ‰ and remains low within the next 0.7 m (average $\delta^{13}C_{TOC}$ = -28.02 ±0.19 ‰, n=8). At 3.9 m of section, $\delta^{13}C_{TOC}$ values increase to 26.95 ±0.16 ‰ (n=37) and remain constant (-26.5 ‰ to -27.3 ‰) for the next 6.5 m before they decrease again to -28.1 ‰ (at 10.6 m) and reach values as low as -28.3 ‰ (at 12.4 m). Between 12.4 m and 12.9 m $\delta^{13}C_{TOC}$ increases to a value of -26.2 ‰ (at 12.9 m) and attains an average of $\delta^{13}C_{TOC}$ = -26.20 ±0.43 ‰ (n=18, 12.9 to 15.8 m), very similar to the pre-excursion $\delta^{13}C_{TOC}$ values in the 0-2.9 m part of the section.

Overall, the $\delta^{13}C_{TOC}$ data show four conspicuous shifts across the record (Fig. 2b): The basal shift at 3.0 m of section to lower $\delta^{13}C_{TOC}$ values ($\Delta\delta^{13}C_{TOC, \text{ single samples at ~3m}}$ = -1.7 ‰) and the uppermost gradual increase between 12.4 m and 12.9 m back to higher $\delta^{13}C_{TOC}$ values ($\Delta\delta^{13}C_{TOC, 12.4m-12.9m}$ = +2.1 ‰) occur within individual lignite seams. Thus, these two shifts are independent of major lithological changes (Fig. 2). In contrast, the $\delta^{13}C_{TOC}$ shifts at 3.9 m ($\Delta\delta^{13}C_{TOC, 3.9m}$ = +0.9) and between 10.3-10.6 m ($\Delta\delta^{13}C_{TOC, 10.6m}$ = -1.3 ‰) correlate with lithological changes. Most importantly, the remarkably stable $\delta^{13}C_{TOC}$ values in the central part of the section (3.9 to 10.6 m) are restricted to the clastic Interbed 2 deposited under marine conditions. Omitting these marine samples (with %TOC <10 %), indicates that the low $\delta^{13}C_{TOC}$ values smoothly tie between Seam 1 and Seam 2 with an average value of -27.68 ±0.43 ‰, whereas $\delta^{13}C_{TOC}$ values in the shallow marine sediments (Interbed 2) are ~0.7 ‰ higher. Higher $\delta^{13}C_{TOC}$ values in the clastic interbed may be surprising at first as marine organic matter is generally more depleted in $^{13}C$ compared to terrestrial organic matter (e.g. Sluijs and Dickens, 2012). However, given the near coastal setting of the Schöningen locality during the early Paleogene we can expect a significant terrestrial contribution to the TOC content (confirmed by the presence of abundant terrestrial derived pollen; Fig. 5), which would increase the $\delta^{13}C_{TOC}$ values (Sluijs and Dickens, 2012). Interestingly, the average $\delta^{13}C_{TOC}$ value (-27.68 ±0.43 ‰) agrees

well with CIE-related $\delta^{13}C_{TOC}$ values with high (~ 80 %) terrestrial contributions of organic matter (data from IODP site 4A; Sluijs and Dickens, 2012). The low $\delta^{13}C_{TOC}$ values at the base of Interbed 2 (3 samples with ~-28 ‰) may indicate reworking of the underlying peat deposits which is supported by scattered lignite material in the sediments.

Detailed biogeochemical investigations of Seam 1 in a nearby section from Schöningen also showed a gradual depletion in $\delta^{13}C_{TOC}$ as well as in mid- and long-chain *n*-alkanes at the top of Seam 1 (Inglis et al., 2015, 2017). Moreover, the absolute %TOC and $\delta^{13}C_{TOC}$ values (Inglis et al., 2015) are in very good agreement with our newly derived data and reveal a statistically significant (single-tailed t-test p< 0.000003) negative shift in $\delta^{13}C_{TOC}$ values at the top of Seam 1 with $\Delta\delta^{13}C_{TOC}$ = ~ -1.0 ‰ (from average $\delta^{13}C_{TOC}$ = -26.53 ±0.30 ‰ to $\delta^{13}C_{TOC}$ = -27.52 ±0.09 ‰) (Fig. 3a).

We consider the excursion in $\delta^{13}C_{TOC}$ within the lignite seams and thus independent of lithological changes to demark the onset and termination of a CIE in the basal part of the Schöningen Formation. It shows an abrupt decrease ($\Delta\delta^{13}C_{TOC}$ = -1.7 ‰ within 0.08 m of section (2.94 to 3.02 m)), but a more gradual increase of $\delta^{13}C_{TOC}$ values ($\Delta\delta^{13}C_{TOC}$ = +2.1 ‰ within 0.52 m of section (12.41 to 12.93 m)). The major shift in $\delta^{13}C_{TOC}$ occurs entirely within the lignite seam with uniformly %TOC, virtually unaffected by marine influences at this stratigraphic level as indicated by rather abrupt lithological transitions on a cm-dm scale (cf. Supplementary Information SI1) and defined palynological transitions between the terrestrial peat deposits to the marine interbed (cf. section 4.3). Thus, we exclude potential mixing of terrestrial and marine (with generally lower $\delta^{13}C$ values) organic matter to be the main driver for the detected carbon isotope excursion.

The described CIE occurs in the two seams surrounding the clastic interbed that includes a prominent *Apectodinium* acme (Riegel et al., 2012). This indicates significant differences in paleoenvironmental conditions when compared to the majority of marine interbeds of the Schöningen Formation that lack abundant *Apectodinium* spp.. Thus, the detected CIE is very likely associated to carbon cycle pertubation during an early Paleogene hyperthermal. At the present point, however, we cannot unequivocally assign the Schöningen CIE to the PETM or the ETM2.

### 4.2 The magnitude of the CIE in European lignite records

Even though the magnitudes of the CIE vary widely among proxy records across the PETM and to a lower extent across the ETM2 event (cf. McInerney and Wing, 2011; Sluijs and Dickens, 2012), the Schöningen CIE is small given the generally large CIE in bulk organic carbon in terrestrial settings (McInerney and Wing, 2011). Terrestrial CIEs related to the PETM are commonly enhanced by ~ 1-3 ‰ compared to those inferred from marine organic matter (typically in the range of -2 to -3 ‰; e.g. Cramer and Kent, 2005; McInerney and Wing, 2011; Sluijs and Dickens, 2012) and the same may apply for the terrestrial expression of the ETM2. Thus, it may be questioned whether the reduced magnitude of the CIE at Schöningen is unique due to (1) local conditions and can be actually related to the PETM, (2) represents another early Eocene hyperthermal, or (3) is an independent feature of the mid latitudinal European near-coastal environments.

In order to address this question, we compare our $\delta^{13}C_{TOC}$ record and the adjacent Schöningen record of Inglis et al. (2015) with published peat mire records along the paleo-North Sea coast line (Fig. 1), namely the Cobham Lignite (UK) (Collinson

et al., 2003, 2009; Pancost et al., 2007) and the Vasterival section (France) (Garel et al., 2013; Storme et al., 2012) that were both assigned to the PETM. Interestingly, all three lignite deposits share characteristic features (Table 1; Fig. 3 and Fig. 4):

(1) Absolute $\delta^{13}C_{TOC}$ values and the range in $\delta^{13}C_{TOC}$ values (~3.2 ‰) are very similar (Fig. 3 and Table 1);

(2) All three records attain similar minimum $\delta^{13}C_{TOC}$ values during the CIE (-27.5 ‰ to -28.8 ‰), averaging at 28.05 ±0.5 ‰ (Fig. 4);

(3) At the onset of the CIE the magnitude of changes in $\delta^{13}C_{TOC}$ (calculated as the difference between the last pre-CIE value and the first CIE value) ranges only between -1.4 to -1.8 ‰ (Fig. 3);

(4) Magnitudes of the CIE in bulk organic matter calculated as the difference between the mean pre-CIE and the mean CIE values range from -0.9 to -1.6 ‰. CIEs calculated as the difference between the mean pre-CIE values and the most negative value during CIE (following McInerney and Wing (2011), cf. Fig. 4) yields magnitudes of -1.1 to -2.3 ‰ (Table 1).

Even though suggestive, these similarities do not provide unequivocal evidence that the Schöningen CIE reflects the PETM; it may well represent another early Eocene CIE. However, the similarity of the CIEs (two of them assigned to the PETM; Collinson et al., 2003, 2009; Pancost et al., 2007; Garel et al., 2013; Storme et al., 2012) between these adjacent lignite records is striking and may argue for a quite uniform behavior of these wetlands during latest Paleocene-early Eocene thermal events.

Overall, the comparison of these geographically adjacent deposits shows that all reported CIE magnitudes of lignite records along the paleo-North Sea are dampened compared to purely continental terrestrial archives but yield a very consistent and robust signal (Fig. 4). Depending on the definition of the CIE, the average magnitude is -1.27 ±0.29 ‰ ("mean-mean") or -1.74 ±0.46 ‰. ("mean-most negative value"). The average decrease of $\delta^{13}C_{TOC}$ values at the onset of the CIE is -1.39 ±0.43 ‰ (Table 1). There are multiple possibilities to explain the dampened magnitude in these deposits such as mixing and dilution of the input signal, occurrence of local signal perturbation (e.g. due to vegetation changes), or differential degradation/preservation of organic matter during the climatic perturbation.

Mixing and dilution of the CIE in the Schöningen estuarine depositional context, where multiple flooding and thus reworking events may have occurred, appears unlikely as the observed CIE onset is sharp (i.e. between 2 samples within 8 cm) and within a lignite seam where no mixing of sediment has been detected (cf. section 4.1). The organic matter of the original peat likely resulted from an ombrotrophic (rain-fed) peat mire (consisting mostly of mosses, ferns, and associated hardwood mire forest, see section 4.3) (Inglis et al., 2015; Riegel et al., 2012) and has to be regarded as autochthonous with transport (if any) only on very short distances, likely meters.

It is possible that the reduced CIE magnitude is a local signal derived by changes in plant communities during the associated hyperthermal. For instance, variable angiosperm : gymnosperm ratios caused significant variations in the recovered $\delta^{13}C_{TOC}$ values of Miocene lignites from Austria (Bechtel et al., 2003) and a similar scenario appears possible at the Paleogene Schöningen locality. However, in our pollen record we do not observe particular changes from angiosperms to gymnosperms

in conjunction with the CIE. Similar to the Schöningen record, the Cobham palynological record lacks major changes in the paleofloral community along with the CIE (Collinson et al., 2003, 2009). Collinson et al. (2009) found only subtle vegetation changes across the PETM in the Cobham lignites and primarily attributed these to changes in the local fire regime. In addition to this, Collinson et al. (2003) also discussed that the carbon isotope variability in the Cobham lignites may have been caused by local changes in the depositional environment, the preservation states, or the plant communities, but at the same time excluded those mechanisms because the major shift in the carbon isotope values occurred without any major lithological or floral changes. These findings are similar to our observations for the Schöningen lignites.

Moreover, marine sediments from the paleo-North Sea exhibit an enhanced CIE of 6-8 ‰ that has been explained by increased terrigenous input (Heilmann-Clausen and Schmitz, 2000; Schmitz et al., 2004; Sluijs and Dickens, 2012). Land plant derived $\delta^{13}C$ values of n-alkanes from two sections of the paleo-North Sea (Denmark) record a decrease of 4–7 ‰ (Schoon et al., 2015), clearly showing that the PETM affected the biosphere around the paleo-North Sea in terms of recording the CIE. However, Schoon et al. (2015) also noted that the differences in the CIE likely arose from local differences in the plant communities or precipitation patterns. Taken together, we think that a local change in vegetation, altering the carbon isotope "input signal" to the peat mires, is unlikely to account for the reduced CIE in the Schöningen lignites.

An alternative scenario is differential degradation/preservation of organic matter in the Schöningen peat mires during a hyperthermal event. Carbon isotope discrimination between litter input and stored (soil) organic matter during degradation/decomposition is governed by fractionation processes during metabolism (typically enriching the residual carbon stock in $^{13}C$) and the selective utilization of compounds (with differing $\delta^{13}C$ values) (e.g. Santruckova et al., 2000). The latter process apparently dominates and can either enhance or suppress a metabolism-related fractionation signal (Santruckova et al., 2000). Minor warming of ~1°C can cause significant increases in carbon respiration rates (on average by 52 % in spring to 60 % in summer; Dorrepaal et al., 2009) in modern high-latitude peatlands. Contemporaneous to the increased respiration rates, an increase in the carbon isotope ratios of the respired $CO_2$ has been interpreted as a shift towards respiration of less $^{13}C$-depleted carbon stocks, likely due to a change in microbial communities.

Even though the paleoenvironmental setting of the Paleogene peat mire clearly differs from the modern high-latitude mires (Dorrepaal et al., 2009), it seems likely that warming related to a hyperthermal generally affected peat mires by increasing respiration rates and causing changes in the microbial community. This could have resulted in specific changes in the $\delta^{13}C$ values of the respired and residual carbon stocks in peatlands. Indeed, Pancost et al. (2007) attributed the shift in $\delta^{13}C$ values of hopanes, a biomarker derived from bacteria, in the Cobham Lignite Beds to an increase in the population of methanotrophic bacteria and, possibly, decreased heterotrophic biomass in this peatland during the PETM. Thus, this study documents a major shift in the microbial community across the PETM (Pancost et al., 2007). Hopane data from Schöningen have been interpreted in terms of a generally mixed methanotrophic and heterotrophic bacterial population (Inglis et al., 2015). The decreasing trend in $\delta^{13}C$ values of hopanes at the top of Seam 1, coinciding with the position of our detected CIE,

may indicate slight changes in the bacterial communities associated with the inferred hyperthermal, even though no significant warming trend based on branched-GDGTs was detected here (Inglis et al., 2015, 2017). Higher-than-expected $\delta^{13}C_{TOC}$ values during the CIE relative to pre-CIE values could result from enhanced microbial degradation processes and changed microbial communities that decompose more $^{13}C$-depleted recalcitrant matter. Generally enhanced respiration rates and changes in the microbial communities due to warming (and wetting) are also likely to appear on a regional scale and are thus, consistent with our finding of generally reduced CIEs in peat mires along the paleo-North Sea.

### 4.3 Environmental changes in the Schöningen peat mire

The repeated change from open estuary/marine to meandering river/peat mire environments is characteristic for the Schöningen Formation and continues even into the middle Eocene Helmstedt Formation (Fig. 1b) (Riegel et al., 2012, 2015). We extended our palynological data set down-section to the underlying Interbed 1 and Main Seam in order to disentangle effects of transgressions/regressions in the coastal setting which were governed by an interplay of eustatic sea level changes, withdrawal of salt towards the salt wall and/or changes in precipitation and subsequent runoff from the direct effects of climate perturbation during the potential hyperthermal associated with the CIE.

During the late Paleocene and early Eocene mire forests, typical for coastal areas along the edge of the paleo-North Sea basin, existed in the area of Schöningen (e.g. Allen, 1982; Willumsen, 2004; Collinson et al., 2009). As inferred from pollen records they consisted essentially of wet swamp forests dominated by Nyssaceae and Cupressaceae s.l. and dryer hardwood mire and background forests characterized by Fagaceae and Myricaceae/Betulaceae (Riegel et al., 2012, 2015). Three groups of palynomorphs can be distinguished (Fig. 5): (1) taxa that occur throughout the entire succession but with frequency maxima in the interbeds 1 and 2 such as pollen of Cupressaceae s.l. (*Inaperturopollenites* spp., *Cupressacidites* sp.), Fagaceae resp. Leguminosae (*Tricolpopollenites liblarensis*), Fagaceae (*Tricolporopollenites cingulum*) or Myricaceae/Betulaceae (*Triporopollenites robustus* group), (2) taxa that are especially abundant in some of the interbed-lignite seam transitions, such as the juglandaceous pollen *Plicapollis pseudoexcelsus* (Main Seam/Interbed 1 and Interbed 1/Seam 1) and *Thomsonipollenites magnificus* (Interbed 2/ Seam 2 and to a lesser extent Main Seam/Interbed 1) (the latter with unknown botanical affinity), and (3) taxa that are strictly confined to lignite seams, e.g. spores of peat mosses (Spagnaceae) or spores of polypodiaceaous ferns.

The separation of samples based on the dominance of one of these three palynomorph groups is reflected by NMDS of the palynomorph data (Fig. 6a). Samples from Seams 1 and 2 plot on the left side of NMDS axis 1 in the ordination space, clearly separated from interbed samples which plot on the right side of the ordination space. This indicates the completely different palynological composition of interbed and lignite samples with dominance of fagaceous and myricaceous/ betulaceous pollen in the interbeds and of spores of peat mosses and polypodiaceous ferns in the lignites (Figs. 6a, b). Samples from the Main Seam plot in the center of the NMDS ordination space indicating that there are slight compositional differences in comparison to Seams 1 and 2. However, the samples are only from the top of the Main seam and may

therefore not reflecting the typical lignite-forming vegetation as recorded in Seams 1 and 2 but more the vegetation of lignite /interbed transitions.

The fagaceous pollen *T. liblarensis* and *T. cingulum* appear to be essentially confined to the CIE (9-14 m in Fig. 5). However, when compared with the older part of the succession (0-6 m in Fig. 5), these taxa appear more frequently in the marine interbeds. Presumably, with the rise of the sea level the respective forests shifted landward and the small wind-transported fagaceous pollen became enriched in the interbeds at this site. For the same reason, pollen of the Cupressaceae s.l. indicative of a swamp forest, occur more frequently in the marine interbeds. Myricaceae/Betulaceae dominated forests as represented by *T. robustus*-group pollen are considered to have grown on better drained, remote mire areas and are therefore less affected by sea-level fluctuations (Riegel et al., 2012).

Floral successions at marine interbed/lignite transitions at Schöningen differ significantly from those of the middle Eocene Helmstedt Formation in nearly lacking pollen of the tropical mangrove elements *Rhizophora*, *Avicennia*, *Nypa* and *Psilodiporites* of unknown botanical affinity (Lenz, 2005; Lenz and Riegel, 2001; Riegel et al., 2012, 2015; this study). Instead, transitions at Schöningen are characterized by *Thomsonipollis magnificus*, *Pistillipollenites mcgregorii* (Fig. 5 and 6), *Plicapollis pseudoexcelsus*, and *Pompeckjoidaepollenites subhercynicus*. The latter two are also known from the middle Eocene of the Helmstedt Formation (Lenz and Riegel, 2001). The absence of tropical mangrove elements, especially *Nypa*, has been interpreted as indicating extratropical conditions during the deposition of the Schöningen Formation in contrast to the true tropical conditions during the middle Eocene (Helmstedt Formation) (Riegel et al., 2012).

Spores of Sphagnaceae (peat mosses) and polypodiaceous ferns are typical lignite related elements throughout much of the Schöningen Formation and often dominate the palynological assemblages (Fig. 5 and 6; Inglis et al., 2015; Riegel and Wilde, 2016; Riegel et al., 2012). The frequent and close association of these spores (*Sphagnumsporites spp., Distancorisporis sp., Tripunctisporis sp., Laevigatosporites spp.*) with charcoal horizons is characteristic for Seam 1 and Seam 2 and has been interpreted as the secondary vegetation succeeding forest fires (Hammer-Schiemann, 1998; Riegel et al., 2012; Inglis et al., 2015; Robson et al., 2015). *Sphagnum* spores sharply decline at the top of Seam 1 and reappear with considerable delay in Seam 2 (above 13.2 m in Fig. 5). This *Sphagnum*-free interval coincides with the range of the detected Schöningen CIE and could potentially reflect a response to the CIE-related climate perturbation. However, a similar distribution pattern of *Sphagnum*-spores has been observed in the lower part of Seam 1 (Fig. 5 and Inglis et al. (2015)) as well as in the Main Seam (Hammer-Schiemann 1998), suggesting that the return of peat mosses is typical for post-fire successions of peat-forming mires following marine incursions at Schöningen. Even though Storme et al. (2012) reported dry/wet cycling across the late Paleocene and early Eocene with rather dry conditions during the main part of the CIE at Vasterival (France), we exclude overall drying as the cause of suppressed proliferation of Sphagnaceae at Schöningen. Similar to the Cobham Lignite (Collinson et al., 2003), waterlogged conditions are indicated at the top of Seam 1 by the presence of freshwater phytoplankton and confirmed by multiple biomarker analyses (Inglis et al., 2015). Regional proxy records indicate increased terrestrial runoff during the warmth of the PETM (Bornemann et al., 2014; Heilmann-Clausen and Schmitz, 2000; Schmitz and Pujalte, 2003), consistent with climate model outputs which show generally increased but also more variable rainfall

during the PETM (e.g. Carmichael et al., 2016, 2017). Therefore, either increased nutrient inputs to the mire and/or climatic changes during the PETM or the subsequent early Eocene hyperthermals may have restrained proliferation of Sphagnaceae and promoted the spread of higher plants (e.g. Cupressaceae s.l. and parent plants of *T. cingulum* and *T. liblarensis*).

Compared to other Paleocene and early Eocene palynological records in the North Sea basin, which indicate significant vegetation changes due to short-lived climate perturbations such as the PETM (e.g. Beerling and Jolley 1998; Eldrett et al. 2014), our palynological data indicate only minor changes of plant taxa associated with the detected CIE. This is confirmed by NMDS, which shows that there are no major changes in the composition of the palynological assemblages when comparing Seam 1 and 2 as well as Interbeds 1 and 2, as both, lignite samples and interbed samples, overlap in the ordination space of the NMDS (Fig. 6b). Furthermore, comparison of post-CIE with pre-CIE (Seam 1) samples as well as comparison of CIE with pre-CIE (Interbed 1) samples reveals no significant differences in the composition of the palynomorph assemblages (Fig. 6c). Therefore, long-term environmental records are needed to identify whether plant community changes were forced by (1) lithological/environmental changes, (2) (hyperthermal-related) climate changes, or (3) a combination of both. Hitherto, changes in plant communities at Schöningen seem to follow natural successions at marine-terrestrial interfaces rather than climatic patterns.

### 4.4 Schöningen in relation to other European lignite records

In order to place our paleoenvironmental observations in a regional framework, we compare the Schöningen record with the nearby lignite sites of Cobham (UK) and Vasterival (F) in which the reported CIEs have been assigned to the PETM, even though we cannot ensure (or exclude) that these records are time-equivalent and represent the same hyperthermal. However, we feel that such a comparison is still valuable in order to detect similar behaviors of these Paleogene wetlands during carbon cycle perturbations.

In the Cobham Lignite record minor qualitative changes among plant species contrast major changes in the composition of plant communities across the PETM onset, which includes the disappearance of ferns and the increase in cupressaceous conifers (Collinson et al., 2003, 2009). This is similar to Schöningen, where the disappearance of ferns prior the onset of the CIE is followed by a similarly high but more fluctuating occurrence of Cupressaceae during the CIE but with similarly high but more fluctuating occurrence of Cupressaceae throughout the record (Fig. 5).

Strikingly similar at both localities, Schöningen and Cobham, is the high abundance of charcoal prior to the detected CIEs. At Cobham, this charcoal is in close association with abundant fern spores and is most likely derived from a secondary vegetation succeeding wildfires at the onset of the PETM (Collinson et al., 2003, 2009). At Schöningen, a high abundance of charcoal occurs in Seam 1 and Seam 2 (Riegel et al., 2012; Robson et al., 2015) with particular high charcoal contents in the upper part of Seam 1 compared to its base (Inglis et al., 2015; Robson et al., 2015). This increase in fire intensity immediately precedes the CIE. Evidence for high frequency of wildfires from Schöningen and Cobham prior to the reported CIEs is compatible with the possibility that the Schöningen CIE could be related to the PETM and the hypothesis that peat burning was an important trigger for the CIE (Kurtz et al., 2003; Moore and Kurtz, 2008).

Another common characteristic of the lignite records at Schöningen, Cobham, and Vasterival is drowning of the peat mires just subsequent to the onset of the detected CIE (Collinson et al., 2003; Garel et al., 2013, this study, Brandes et al., 2012; Riegel et al., 2012). If the CIE in the Schöningen record is associated with the PETM, the global transgressional phase (e.g. Sluijs et al., 2011) likely resulted in the deposition of the marine clastic Interbed 2 at Schöningen during much of the PETM

(Fig. 2). The return of the Schöningen peat mire (Seam 2) may have been caused by a decrease of thermal expansion of the ocean and a concomitant global regression during cessation of the PETM. At the same time, increased sediment supply from the hinterland during the PETM (e.g. Bornemann et al., 2014; Heilmann-Clausen and Schmitz, 2000) filled the available accommodation space. Any subsequent Eocene hyperthermal may have resulted in a similar behavior.

## 5 Summary and Conclusion

Bulk organic carbon isotopic and palynological data from an alternating succession of lignite and clastic deposits in the basal Schöningen Formation (Germany) show a negative CIE ($\Delta\delta^{13}C_{TOC}$ = -1.7 ‰) and an *Apectodinium* acme. Our CIE record highlights that the interval of highest fire frequency in the Schöningen Formation (Seam 1; Robson et al., 2015) is clearly associated with the carbon excursion. These characteristics together with the available age constraints for the formation do not allow for a robust correlation to a particular Paleocene/Eocene hyperthermal, but do not exclude that this CIE is related

to the PETM. Paleofloral changes related to the CIE time interval are minor and most changes follow natural successions. Thus, only long-term environmental records appear suitable to distinguish if plant communities changed due to (1) lithological/environmental changes, (2) climate change associated with the CIE, or (3) a combination of both.

Tentative comparison with the $\delta^{13}C_{TOC}$ records of nearby peat mire records along the paleo-North Sea coast line (Cobham, UK; Vasterival, F) that have been associated with the PETM shows that the carbon isotopic composition of these lignites

yields a reduced, compared to marine or other terrestrial archives, but consistent CIE with a magnitude of ~ -1.3 ‰ that appears to be a robust regional signal along coastal sites of the paleo-North Sea.

Common features of the Schöningen and the Cobham Lignite records emerge, such as a similar CIE, similar paleo-floral successions, and drowning of peat mires during the major body of the CIE. Furthermore, both records yield evidence of increased fire activity such as increased charcoal contents in combination with the appearance of ferns and peat mosses prior

to the CIE. Overall, the similarities between these Paleogene-Eocene European wetland records are striking and it can be hypothesized that they either represent the same hyperthermal or that different Paleocene-early Eocene carbon cycle perturbations and associated hyperthermals had similar effects on mid-latitudinal wetlands along the paleo-North Sea.

## Acknowledgements

O.L. acknowledges support through DFG LE 2376/4-1. We further thank Karin Schmidt for valuable field support and Jens

Fiebig, Sven Hofmann and Ulrich Treffert for technical assistance. Gordon Inglis kindly supplied his original $\delta^{13}C$ dataset

for the top of Seam 1. We are grateful to the Helmstedter Revier of MIBRAG (formerly BKB and later EoN) for access to the sections and technical assistance in the field. We thank Joost Frieling and an anonymous referee for their insightful reviews as well as Carlos Jaramillo and Gerald Dickens for their comments that helped to significantly improve this manuscript.

**Data/Sample availability**

All shown and discussed data is available in the Supplementary Information. Samples are stored in the Senckenberg collections and are available upon request.

**Author contributions**

W.R., V.W. and A.M. designed the study. K.M. composed the paper. K.M. and A.M. conducted the geochemical analyses
and evaluated the results. W.R., V.W., and O.L. provided sample material, regional geological expertise and conducted palynological analyses. All authors edited the final version of this manuscript.

**Competing interests**

The authors declare that they have no conflict of interest.

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

**Table**

**Table 1: Compilation of $\delta^{13}C_{TOC}$ data sets from European lignite deposits.**

| Record | min. and max. $\delta^{13}C$ value of each record [‰] | $\delta^{13}C$ range [‰] | CIE onset[b] [‰] | CIE magnitude ("mean-mean")[c] [‰] | CIE magnitude ("mean-most negative")[d] [‰] | Reference |
|---|---|---|---|---|---|---|
| Schöningen (D) | -25.06 to -28.29 | 3.23 | -1.66 | -1.26 | -1.53 | this study |
| Schöningen (D) | -25.95 to -27.65 | 1.70 | -0.68 | -0.92 | -1.11 | Inglis et al., 2015 |
| Cobham (UK) | -24.47 to -27.50 | 3.03 | -1.40 | -1.60 (-1.16)[e] | -1.97 (-1.53)[e] | Collinson et al., 2003 |
| Vasterival (F) | -25.4 to -28.8[a] | 3.4 | -1.8 | -1.5 | -2.3 | Storme et al., 2012 |

[a] exact data not given in the paper, manually extracted from the published figure.

[b] CIE onset is calculated as the difference between the last pre-CIE and the first CIE sample.

[c] CIE magnitude calculated as the difference between the mean pre-CIE and the mean CIE value.

[d] CIE magnitude calculated as the difference between the mean pre-CIE and the most negative value of the CIE (following McInerney and Wing (2011)).

[e] CIE magnitude omitting the described increase of $\delta^{13}C_{TOC}$ at the basal part of the Cobham record (Collinson et al., 2003) and only taking the last 6 samples prior to the onset of the CIE as reference.

**Figure**

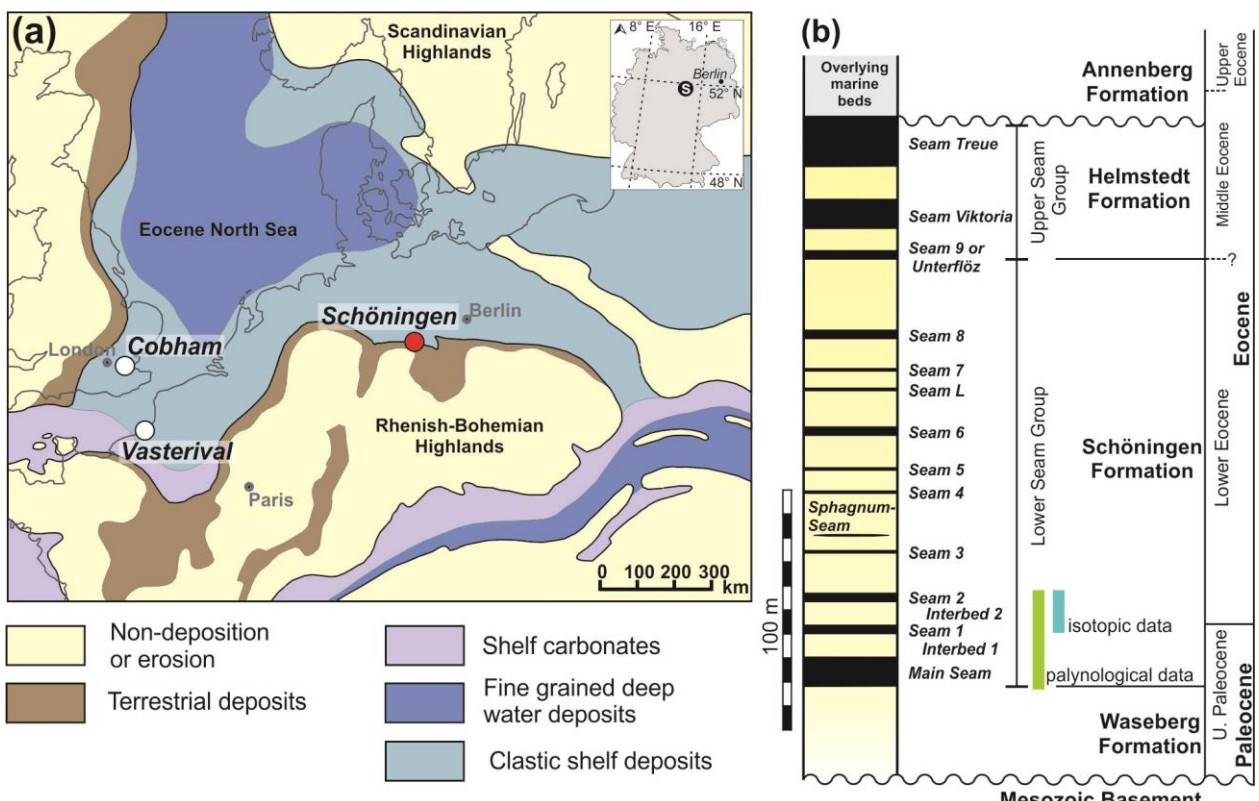

**Figure 1: (a) Palaeogeographic map of northwestern Europe during the early Eocene (adapted from Ziegler, 1990) showing the locations of the Schöningen open cast mine (D), Cobham (UK) and Vasterival (F). (b) Schematic stratigraphy of the Schöningen area (adapted from Brandes et al., 2012).**

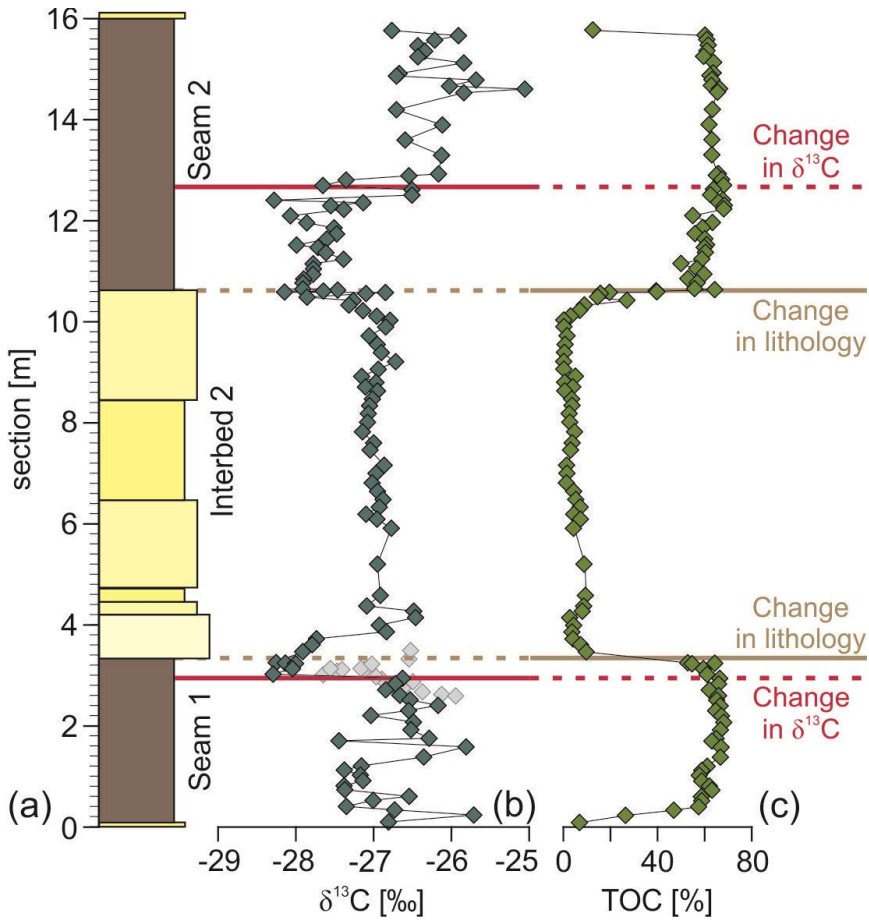

**Figure 2: (a) Composite stratigraphic section, (b) δ¹³C values of bulk organic matter, and (c) total organic carbon (%TOC). Lines denote lithological changes between the marine interbed and the lignite seams (brown) and changes in δ¹³C values independent of lithological changes (red).**

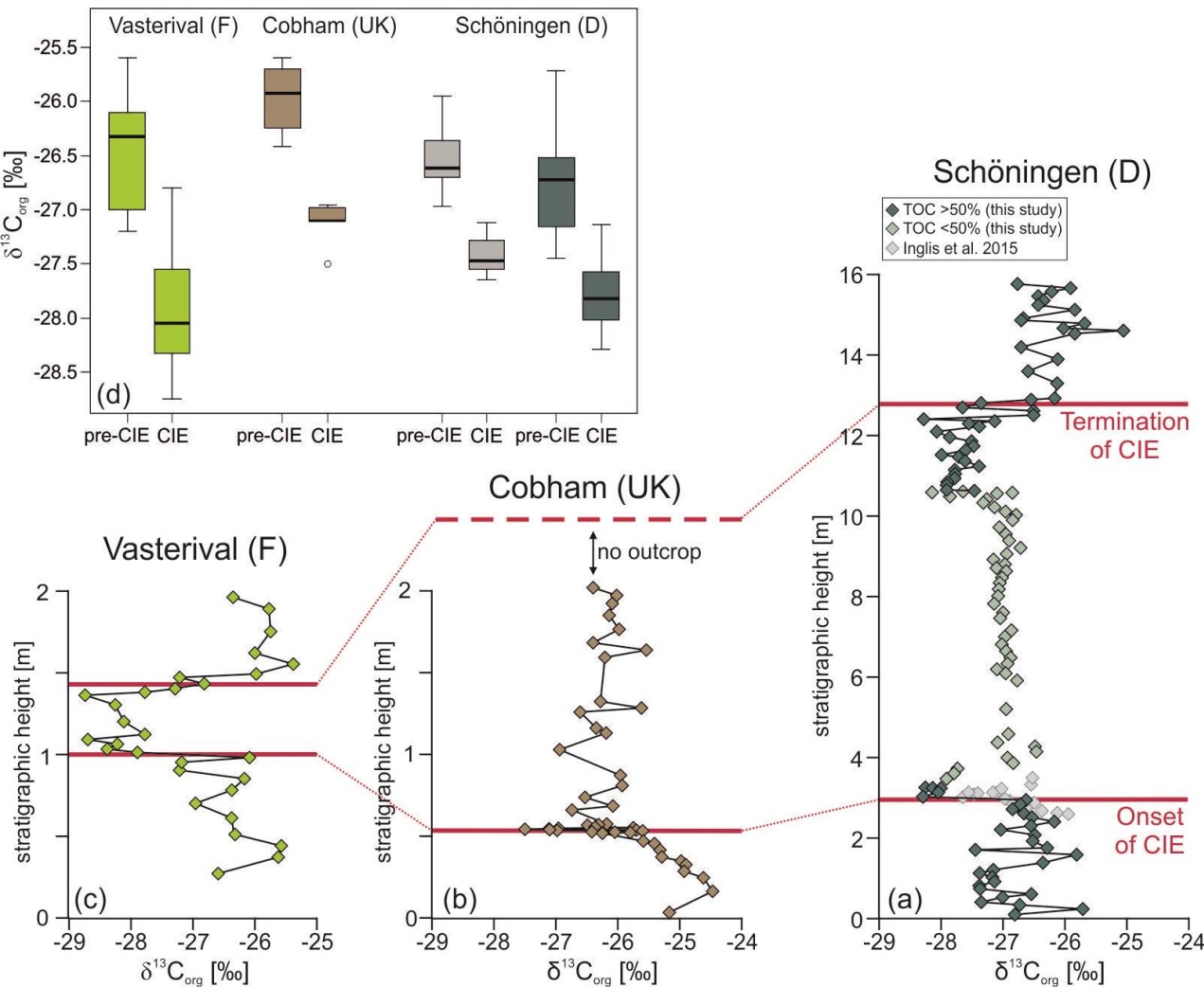

**Figure 3: Comparison of mid-latitudinal wetland δ¹³C_TOC records surrounding the paleo-North Sea (cf. Fig. 1): (a) Schöningen, Germany (this study (green symbols), Inglis et al. (2015) (grey symbols)), (b) Cobham, UK (Collinson et al., 2003), (c) Vasterival, France (Storme et al., 2012). Note the different stratigraphic thicknesses due to different sediment accumulation and preservation conditions in the individual depositional environments. (d) Carbon isotopic differences at the onset of the CIE.**

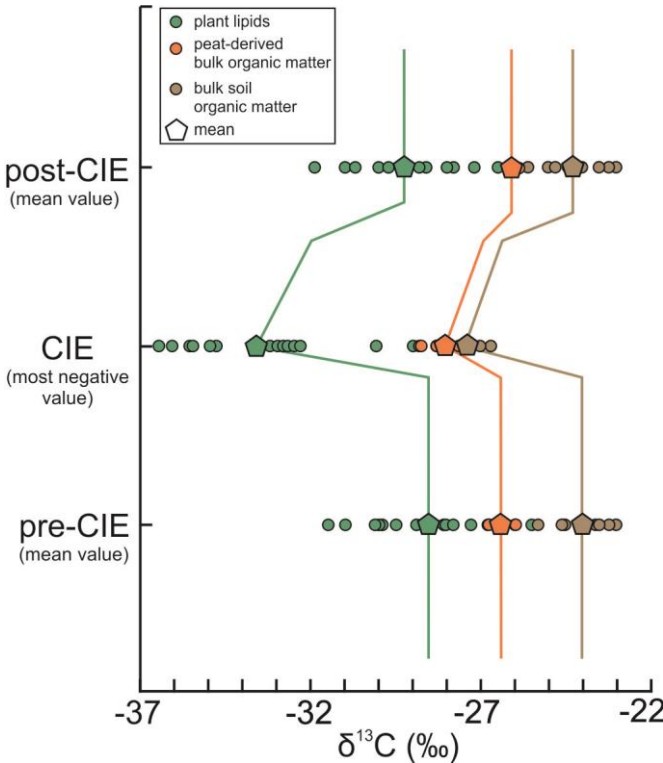

**Figure 4: Comparison of terrestrial carbon isotope excursions (CIE). Data from plant-derived lipids (yellow) and soil organic matter (green) are from McInerney and Wing (2011). Data from peat-derived bulk organic matter are compiled from new (this study) and published studies (Collinson et al., 2003; Inglis et al., 2015; Storme et al., 2012). Filled symbols represent mean values. Whereas the pre- and post-CIE values represent mean $\delta^{13}$C values, the CIE is given as the most negative values, following McInerney and Wing (2011).**

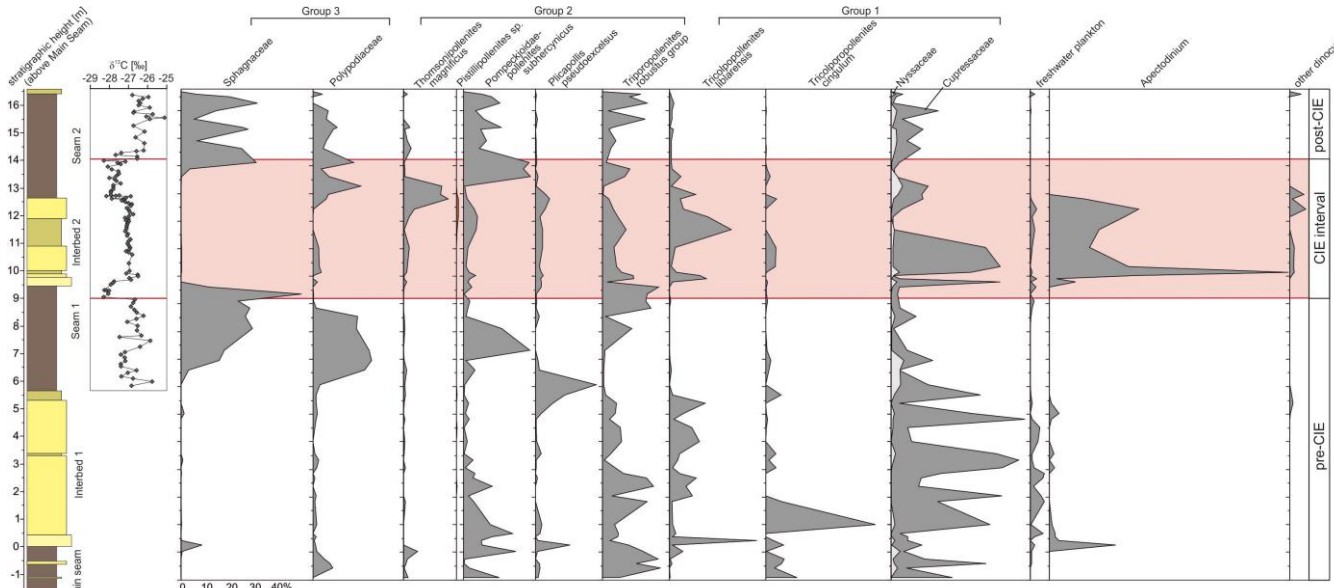

**Figure 5: Simplified pollen diagram, comprising the top of the Main Seam up to the top of Seam 2, showing frequency and palynological abundance changes between pre-, peak- and post-CIE intervals. The carbon isotope data is shown for comparison. Due to the different thicknesses of the sections (cf. Fig. 2), it has been tied to top and base of Seam 1 and top and base of Seam 2.**
**5  Red bar indicate the carbon isotope excursion. The palynological record is characterized by three palynological groups: (1) elements with frequency maxima in the clastic interbeds (representing the hinterland), (2) interbed-lignite seam transitional elements and back mangrove equivalents, and (3) elements dominating the lignite seams.**

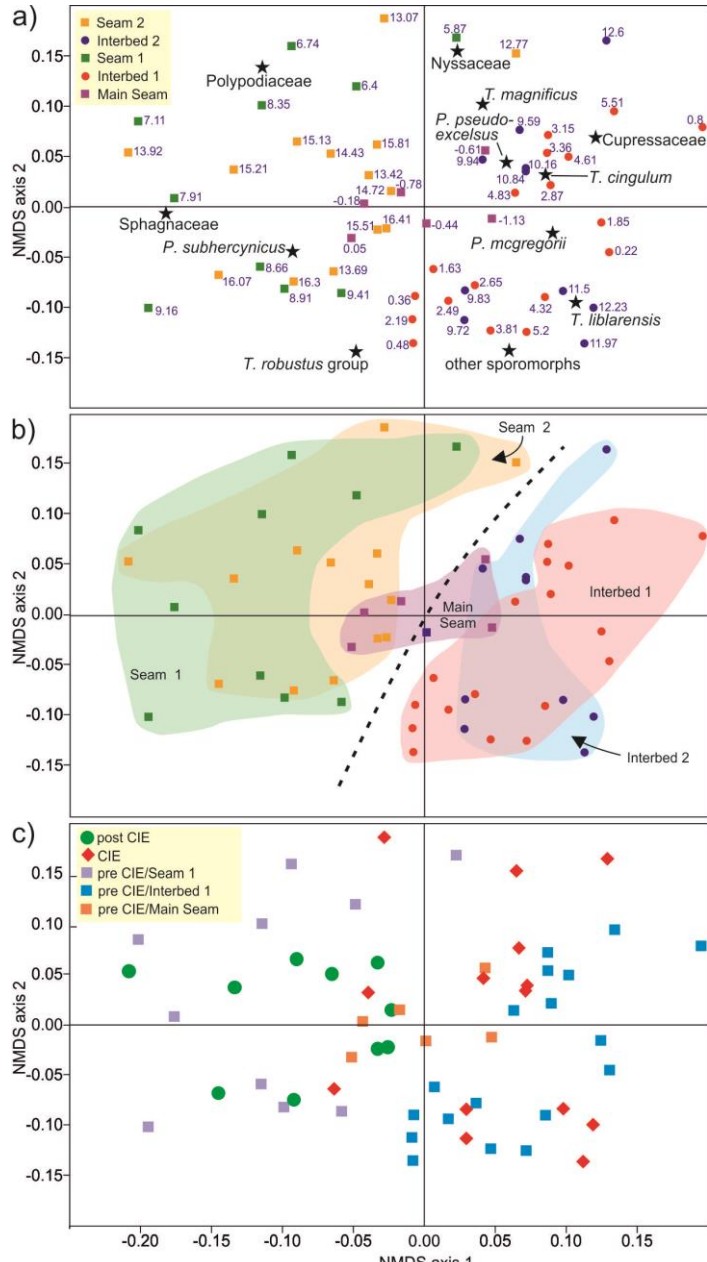

**Figure 6: Non-metric multidimensional scaling (NMDS) of palynological data from the Schöningen Formation, comprising 59 samples from the top of the Main Seam up to the top of Seam 2, using the Bray-Curtis dissimilarity and the Wisconsin double standardized raw data values. (a) Scatter plot of the first two axes showing the arrangement of samples and taxa. The different symbols represent samples of the different lithological units. (b) Scatter plot of the first two axes showing the arrangement of samples with frames that indicate different lithological units in the succession. (c) Scatter plot of the first two axes showing the arrangement of samples separated by different symbols in pre-, peak-, and post-CIE samples.**