# Peer review of "Palaeoenvironmental response of mid-latitudinal wetlands to Paleocene-Early Eocene climate change (Schöningen lignite deposits, Germany)"

_Climate of the Past, 2019_

## Short Comment (SC1) · 28 Feb 2019

**Carlos Jaramillo**

jaramillo.pollen@gmail.com

Received and published: 28 February 2019

This manuscript presents an interesting dataset, but there several problems that need to be solved before it can be published, including: 1. Raw palynological counts must be presented (rather than percentages) 2. Because D13C values of bulk sediments can be affected by the total organic carbon of a sample (Wing et al Science 310, 993, 2005), the Wing residuals method (Wing et al Science 310, 993, 2005), the Wing residuals method (Wing et al Science 310, 993, 2005) needs to be used rather than the actual bulk value. In other words, the residual rather than the bulk needs to be plotted against the stratigraphy. I did a plot of TOC and D13C, see attached, and there seems to be a TOC-D13C correlation at both high and low TOC

values. 3. Something is not quite right about the PETM thickness. There are 180 meters for the Lower Eocene in the section (fig 2, and fig 3 of Brandes et al paper). Early Eocene spans 8.2 my, that would give 45Ky per meter, and 10 meters would be about half a million years, that is twice to three times the span of the PETM. Brandes did a basin modeling analysis and ages for the modeling were derived from "interpolated from literature data" (Table 2). And the "literature data: in the paper refers to "According to these sources, a sample from slightly above the Main Seam has been assigned to dinoflagellate zone D5b (Ahrendt et al., 1995), the base of which is dated to about 54.8 my b.p. (Kothe, 2003). The latter age may thus serve as an approximation for the base of the Main Seam (Gurs et al. ,2002) and the Schoningen Formation. A radiometric age of 46 my b.p. and dinoflagellates indicative of zone D9na have been derived from slightly below the Heidberg unit by Ahrendt et al. (1995) and Lietzow Ritzkowski (1996, 2005a), but a discrepancy is caused by the fact that according to Kothe (2003), dinoflagellate zone D9a ranges from 50.51 to 48.5 my b.p". It seems that 1) the Brandes modeling is not independent as it relies on the ages, and 2) ages themselves have discrepancies. If zone D5b is slightly above Main seam, it would imply for the PETM to be below, not above the Main Seam. The most difficult task when working the PETM is having strong evidence of the precise stratigraphic location of the PETM. It should be fully demonstrated. This manuscript requires a much stronger argument to support the position of the PETM. For instance, it would very useful to have the D13C1 record of the entire early Eocene to see if the long-term negative excursion of the EETM is recorded. 4. A simple inspection of the pollen data is not enough. The palynological analysis requires a statistical test to support the conclusion that there is little change across the PETM. For example, a multidimensional scaling could be run using the Chao-dissimilarity index to test if the differences within PETM samples are as high as differences among pre, post and PETM samples.

Fig. 1. toc vs d13C

СЗ

---

## Short Comment (SC2) · 19 Mar 2019

Came across this manuscript randomly, and thought to make a few comments. Hopefully constructive and useful.

Jerry Dickens

– Page 1 – Lines 8-9: There remains zero evidence that the massive carbon input caused the warming during the PETM. Rather, all evidence suggests a coupled system where warming and carbon input are linked in positive feedback. Rewrite, especially as it ultimately relates to the MS content. – Page 2 – Line 17: Change "shelfs" to

"continental slopes". Phase boundary constraints do not allow gas hydrate to occur on continental shelves. Lines 20-21. This should be reworded. As highlighted by Sluijs and Dickens (2012), there is a major difference between the local expression of the global CIE across the PETM, and what occurred to exogenic carbon cycle (where mass balance comes into discussion).

Line 27: Importantly note that coastal wetlands are generally a location of transient carbon storage on the time scale relevant to the PETM. Carbon is generally not buried in these environments on the >10,000 year time scale, especially when considering changes in sea level, although there are obviously some good examples (aka the present-work). The current writing sort of recognizes this – it's the word "sink" that is problematic. – Page 3– **Lines 3-16: It should be noted that spikes in Apectodinium abundance occur during other hyperthermals of the early Paleogene. The key to the PETM is that there is a special marker species (A. augustum) that seems to have existed only during the PETM. Without documenting this dinocyst, the stratigraphic record shown could be one of the proximal hyperthermals (e.g., H1/ETM2).

– Page 5– **Lines 30-31: As above, it has not been demonstrated that this horizon is, in fact, the PETM. Importantly, though, at one level it does not detract from the MS significance, because there is a growing view that the main hyperthermals are related and have similar systemtic repsonses.

If it cannot be conclusively proven that the interval is the PETM, this should be stated. There should then be some key rewriting and an additional paragraph that notes that much is written with an assumption, but most of the basics would also apply if the interval was instead another lesser but major early Paleogene hyperthermal.

Starting about Line 18, the reading gets awkward, because not always clear what constitutes a paragraph. – Page 6– Lines 15-16: Note that the good and intriguing ideas presented by Trampush and Hajek do not really apply to most deep-sea records, from which the duration has been estimated.

**Line 24+: This is interesting, but all hinges on the correct stratigraphy, something that has to be bolstered better.

– Page 11 – Line 2: Okay, except then, the prominent CIE was not caused by the burning of peat. Overall, this is important and relates to the initial comment above – the CIE and associated massive carbon input was very likely mostly/partly a response to major changes in Earth systems not

---

## Short Comment (SC3) · 19 Mar 2019

Apologies, but things got scrambled in .pdf conversion. I now also see that Carlos Jaramillo gave commentary. His main critical point is the same as mine: basically, how do you know you are looking at PETM? Recent papers by Slotnick et al. (NZJGG, 2015), Laurentano et al. (COP, 2015), Luciani et al. (COP, 2016) and Westerhold et al. (Paleo. Paleo. 2018) really highlight this issue. The MS should very much be rewritten to address this, although as noted, a very good paper can emerge irrespective.

Jerry

– Page 1 – Lines 8-9: There remains zero evidence that the massive carbon input caused the warming during the PETM. Rather, all evidence suggests a coupled system where warming and carbon input are linked in positive feedback. Rewrite, especially as it ultimately relates to the MS content. – Page 2 – Line 17: Change "shelfs" to "continental slopes". Phase boundary constraints do not allow gas hydrate to occur on continental shelves. Lines 20-21. This should be reworded. As highlighted by Sluijs and Dickens (2012), there is a major difference between the local expression of the global CIE across the PETM, and what occurred to exogenic carbon cycle (where mass balance comes into discussion).

Line 27: Importantly note that coastal wetlands are generally a location of transient carbon storage on the time scale relevant to the PETM. Carbon is generally not buried in these environments on the >10,000 year time scale, especially when considering changes in sea level, although there are obviously some good examples (aka the present-work). The current writing sort of recognizes this – it's the word "sink" that is problematic. – Page 3– **Lines 3-16: It should be noted that spikes in Apectodinium abundance occur during other hyperthermals of the early Paleogene. The key to the PETM is that there is a special marker species (A. augustum) that seems to have existed only during the PETM. Without documenting this dinocyst, the stratigraphic record shown could be one of the proximal hyperthermals (e.g., HI/ETM2).

– Page 5– **Lines 30-31: As above, it has not been demonstrated that this horizon is, in fact, the PETM. Importantly, though, at one level it does not detract from the MS significance, because there is a growing view that the main hyperthermals are related and have similar systemtic repsonses.

If it cannot be conclusively proven that the interval is the PETM, this should be stated. There should then be some key rewriting and an additional paragraph that notes that much is written with an assumption, but most of the basics would also apply if the interval was instead another lesser but major early Paleogene hyperthermal.

Starting about Line 18, the reading gets awkward, because not always clear what constitutes a paragraph. – Page 6– Lines 15-16: Note that the good and intriguing ideas presented by Trampush and Hajek do not really apply to most deep-sea records, from which the duration has been estimated.

\*\*Line 24+: This is interesting, but all hinges on the correct stratigraphy, something that has to be bolstered better.

– Page 11 – Line 2: Okay, except then, the prominent CIE was not caused by the burning of peat. Overall, this is important and relates to the initial comment above – the CIE and associated massive carbon input was very likely mostly/partly a response to major changes in Earth systems not the primary driver.

———————————————————

---

## Referee Comment (RC1) · Anonymous Referee #1 · 22 Mar 2019

The manuscript by Methner et al. present new interesting datasets that represent a great contribution to the available terrestrial records from Eocene sections. The authors present new d13C_TOC and TOC (%) data, as well as palynological data from the mid-latitude coastal site of Schoningen. The main conclusion is the identification of the PETM in the CIE represented by the onset of a negative excursion in d13C TOC in Seam 1 and a positive shift in Seam 2, as well as the presence of Apectodium in the marine interbed comprised between the two lignite seams.

The paper is well-written and properly structured and addresses interesting scientific questions which fit the scope of publications in Climate of the Past. Figures and tables

are of good quality and relevant to the manuscript. However, I do recommend to rethink and reframe the manuscript because of the following problems.

1) The identification of the PETM. My main concern is that the evidence brought forward to define the CIE as the PETM should be discussed with care. The authors briefly acknowledge that the identification of the PETM in this interval has been suggested before by Riegel et al., (2012) but fail to discuss the possible pitfalls of this assumption: a) Ages: possible age discrepancies in the dating of the section should be discussed in the manuscript. I refer to the detailed comment by Carlos Jaramillo, who has already noted that "independent" dating by Brandes et al (2012) is relying on ages related to the dinoflagellate zones. This should be addressed in a section of the manuscript b) Thickness: a 10-m thick CIE; this requires a very high sedimentation rate (as noted in line 13) but is this reasonable? How is this changing from one lithology to the other in the transition from lignite to the marine interseam? Also, if we accept an almost linear sedimentation rate ($\sim$0.5 Myr/m) for the whole section, then the duration of the CIE doubles. How can you exclude that this is not the case? And how can you exclude that the CIE is not, for example, the expression of ETM2/H2 hyperthermals (which would together last about 400 kyr)? c) Hyperthermals: the acme of the Apectodium is the strongest evidence use to correlated this interval with the PETM. This is quite a compelling evidence, however, as noted by Jerry Dickens in his comment, Apectodinium augustum is the diagnostic species found in PETM section. Is there evidence for its presence?

In general, I recommend incorporating those points in the discussion, discussing more in detail all the possible pitfalls associated with a univocal identification of the PETM as well as the strong evidence in support of it. I also suggest following Carlos Jaramillo's comments on the raw palynological counts and the TOC vs d13C TOC correlation plot.

Minor comments: Page 1 Lines 6-9: The release of isotopically light carbon was most likely a feedback of the long-term warming rather than the cause. Rephrase. Line 21-23: another problem associated with the interpretation of sources and mechanisms is

local signals in different records Page 2 line 9: A. augustum Page 5: line30 to Page 6 line 16: this part should be rewritten with more care, in the view of the main comment about the definition of this interval as the PETM Page 6 line 12: add a reference here for the PETM duration

———————————————————

---

## Referee Comment (RC2) · Joost Frieling (Referee) · 23 Apr 2019

Please see attached pdf for comments

Please also note the supplement to this comment:
https://www.clim-past-discuss.net/cp-2019-20/cp-2019-20-RC2-supplement.pdf

---

## Author Comment (AC1) · 30 Jun 2019

**Comment 2**

**Gerald Dickens,** jerry@rice.edu,

Apologies, but things got scrambled in .pdf conversion. I now also see that Carlos Jaramillo gave commentary. His main critical point is the same as mine: basically, how do you know you are looking at PETM? Recent papers by Slotnick et al. (NZJGG, 2015), Laurentano et al. (COP, 2015), Luciani et al. (COP, 2016) and Westerhold et al. (Paleo. Paleo. 2018) really highlight this issue. The MS should very much be rewritten to address this, although as noted, a very good paper can emerge irrespective.

Jerry

Dear Jerry,

Thank you very much for your interest in our study and your insightful and helpful comments. We really appreciated this and hope that the revised manuscript meets your expectations.

Katharina

– Page 1 – Lines 8-9: There remains zero evidence that the massive carbon input caused the warming during the PETM. Rather, all evidence suggests a coupled system where warming and carbon input are linked in positive feedback. Rewrite, especially as it ultimately relates to the MS content.

- Agreed. We acknowledge that this wording is clearly misleading and shouldn't appear in the abstract or the manuscript. As a result of major rewriting of the manuscript, the abstract has been changed significantly and this part has been deleted from the abstract.
- In addition, we carefully checked the remaining manuscript for this misleading causality, but couldn't detect any further records.

– Page 2 – Line 17: Change "shelfs" to "continental slopes". Phase boundary constraints do not allow gas hydrate to occur on continental shelves.

- We followed the suggestion of the reviewer and changed the wording accordingly. (p. 2, ln. 20)

Lines 20-21. This should be reworded. As highlighted by Sluijs and Dickens (2012), there is a major difference between the local expression of the global CIE across the PETM, and what occurred to exogenic carbon cycle (where mass balance comes into discussion).

- We acknowledge this comment and extended this paragraph in order to be more careful to distinguish local preserved CIEs vs. the global carbon cycle perturbation. (p. 2, ln. 24-29). This adds to a major part of the review #2, being concerned about carbon signal mixing and the local vs. global significance of the record to the global CIE that we address in section 4.1.

Line 27: Importantly note that coastal wetlands are generally a location of transient carbon storage on the time scale relevant to the PETM. Carbon is generally not buried in these environments on the >10,000 year time scale, especially when considering changes in sea

level, although there are obviously some good examples (aka the present-work). The current writing sort of recognizes this – it's the word "sink" that is problematic.

- We agree with the comment that coastal peatland can act as both, sources and sinks of carbon on different timescales. We followed the suggestion of the reviewer and changed the wording to dampen the emphasis of wetlands being a carbon sink. (p. 2, ln. 32 – p. 3, ln. 3)

– Page 3– **Lines 3-16: It should be noted that spikes in Apectodinium abundance occur during other hyperthermals of the early Paleogene. The key to the PETM is that there is a special marker species (A. augustum) that seems to have existed only during the PETM. Without documenting this dinocyst, the stratigraphic record shown could be one of the proximal hyperthermals (e.g., HI/ETM2).

- Please see reply below.

– Page 5– **Lines 30-31: As above, it has not been demonstrated that this horizon is, in fact, the PETM. Importantly, though, at one level it does not detract from the MS significance, because there is a growing view that the main hyperthermals are related and have similar systemtic repsonses. If it cannot be conclusively proven that the interval is the PETM, this should be stated. There should then be some key rewriting and an additional paragraph that notes that much is written with an assumption, but most of the basics would also apply if the interval was instead another lesser but major early Paleogene hyperthermal.

- This is a major issue that has been similarly addressed by all reviews. We acknowledge that in the first version of the manuscript a clear assignment of our detected CIE to the PETM might have been too bold.
- We acknowledge this by
  - (1) adding a whole new section (now section 2) to the manuscript in order to describe the available age constraints and the pitfalls/discrepancies of them in more detail (section 2; p. 3, ln. 13 – p. 4, ln. 10)
  - (2) being more careful in our wording while describing solely the CIE and not unequivocally relating the CIE to the PETM, ETM2 or any other Early Eocene hyperthermal (section 4.1; p. 7, ln. 20-24).
  - (3) discussing tentatively the possible assignment of the CIE to the PETM vs. any other Early Eocene hyperthermal (section 4.2; p. 8, ln. 16-20)
- We rephrased major parts of section 3.1, now section 4.1 (p. 6/7) and included a statement that we compare the European wetland records despite the possibility that they may reflect different hyperthermal events in section 4.2 (p. 7, ln. 31 – p.8, ln. 6) and 4.4 (p. 12, ln. 25-29).
- In order to place our results in a more regional framework, we still perform the comparison with nearby lignite sites (Cobham, Vasterival) in which the reported CIEs have been assigned to the PETM. We feel, in agreement with this comment, that this comparison might still be valuable to detect similar behaviors of these Paleogene wetlands during carbon cycle perturbations. However, we now clearly state that there is no sufficient proof that these records are time-equivalent as they all have their limitations when it comes to age assignment (in section 4.1, 4.2 and 4.4).

Starting about Line 18, the reading gets awkward, because not always clear what constitutes a paragraph.

- We appreciate this comment and checked the formatting of the manuscript as well as the composition of the paragraphs to provide better clarity/readability to the reader. Adding to the comment and reply above, this paragraph has been largely rewritten (p. 6/7).

– Page 6– Lines 15-16: Note that the good and intriguing ideas presented by Trampush and Hajek do not really apply to most deep-sea records, from which the duration has been estimated.

- We appreciate this comment very much. We use this study to demonstrate that each record, especially terrestrial records, can be subject to large variability in proxy signal recording. However, as the available age constraints of the Schöningen Formation, also used by Brandes et al. (2012) to deduce sedimentation rates, are currently rather weak and a major change of the manuscript is the description of the CIE rather than its assignment to the PETM, we do not feel that calculation of the event duration in such an environment can be robust. Therefore, we deleted parts of this paragraph.

**Line 24+: This is interesting, but all hinges on the correct stratigraphy, something that has to be bolstered better.

- We address the possibility that the detected CIE is related to the PETM, but similarly to any other Early Eocene hyperthermal in an open way to the reader (p. 7, ln. 20-24; p. 7, ln. 31 – p.8, ln. 6; p. 8, ln. 16-20; p. 12, ln. 25-29).
- As stated above, we – despite assessing the CIE in a more open way, not relating it to a particular hyperthermal – feel that it is still valid and useful to make such a regional comparison with other nearby Early Eocene lignite records. To provide clarity to the reader about this, we included short statements in section 4.2 (p. 7, ln. 31 – p.8, ln. 6 and p. 8, ln. 16-20) and section 4.4. (p. 12, ln. 25-29)
- The comparison of the detected CIE with the reported PETM-related CIEs (Cobham, Vasterival) reveals some striking similarities between these records. As a consequence, one can hypothesize that these records reflect either the same hyperthermal event (presumably the PETM as the Cobham and the Vasterival records are assigned to it), or, alternatively, Early Eocene carbon cycle perturbations/hyperthermal events affecting wetlands along the paleo-North Sea in a similar way.
- In regard to this, we are aware of a third hypothesis namely that all three CIE records reflect the same hyperthermal that is NOT the PETM. However, we try to be as careful as possible without judging upon pitfalls of other studies.
- Overall, we feel in agreement with this comment that this comparison might still be valuable to detect similar behaviors of these Paleogene wetlands during carbon cycle perturbations.

– Page 11 – Line 2: Okay, except then, the prominent CIE was not caused by the burning of peat. Overall, this is important and relates to the initial comment above – the CIE and associated massive carbon input was very likely mostly/partly a response to major changes in Earth systems not the primary driver.

- We agree with this comment and try to be careful in phrasing this relationship.

---

## Author Comment (AC2) · 30 Jun 2019

**Comment 1**

Carlos Jaramillo, jaramillo.pollen@gmail.com

Dear Carlos,

Thank you very much for your interest in our study and your thoughtful comments on our manuscript. We hope that we could meet your expectations in the revised manuscript.

Katharina

This manuscript presents an interesting dataset, but there several problems that need to be solved before it can be published, including:

1. Raw palynological counts must be presented (rather than percentages)

- We acknowledge that these data should be presented and follow this suggestion. The raw palynological counts can now be found in the Appendix (section SI3, table SI3).

2. Because D13C values of bulk sediments can be affected by the total organic carbon of a sample (Wing et al Science 310, 993, 2005), the Wing residuals method (Wing et al Science 310, 993, 2005) needs to be used rather than the actual bulk value. In other words, the residual rather than the bulk needs to be plotted against the stratigraphy. I did a plot of TOC and D13C, see attached, and there seems to be a TOC-D13C correlation at both high and low TOC values.

[Figure]

- We appreciate this comment as this method is a valid approach for carbon isotope ratios of soil TOC and %TOC (Wing et al., 2005) and very interesting to apply to our dataset.

- In brief, we find that a **direct assignment of this approach is not possible due to the very different nature of the two settings** (lignite seams/marine sediments and mud-rock paleosols). We deduced %TOC vs. $\delta^{13}C_{TOC}$ relationships from our data set and found very different results to Wing et al. (2005). In particular, we found no relationship across the whole data set ($R^2$ = 0.002), a moderate positive linear relationship for samples with %TOC >50% (lignite seams) ($R^2$ = 0.208), and again no relationship for samples with %TOC <50% (marine interbed) ($R^2$ = 0.039). If we nevertheless apply these "transfer functions" (for details see below), we **maintain the deduced negative carbon isotope excursion**.
- In summary, we feel that exploring on such a relationship in lignite samples and/or modern peat samples would go beyond the scope of this paper, but consider this an interesting approach that needs further investigation.

  In detail:
- Wing et al. (2005) found a strong negative exponential relationship between %TOC and $\delta^{13}C_{TOC}$ in mud-rock paleosol samples in the sub-groups of PETM ($R^2$ = 0.623) and Paleocene-Eocene ($R^2$ = 0.618) samples. For comparison, modern soil samples show a coefficient of determination of 0.952.
- Plotting our data (with the %TOC as the independent variable and $\delta^{13}C_{org}$ as the dependent variable), **no such relationship could be determined** (see plot below). The coefficient of determination ($R^2$) is 0.002 and a F-test revealed no statistical relationship between these two variables.
- Considering only lignite values with %TOC > 50% yields a moderate positive linear relationship ($R^2$ = 0.208) that is statistically significant (F-test). Considering values of %TOC < 50% (marine interbed, with typical %TOC <10%), we found a weak negative linear relationship ($R^2$ = 0.039) that is statistically not significant (F-test).
- Applying the deduced regressions (as transfer functions) -*even though the second regression is not significant*- to our data set in order to plot the residuals from expected $\delta^{13}C_{TOC}$ values, **maintains the deduced negative carbon isotope excursion**. The magnitude is slightly reduced at the CIE onset with -1.22 ‰ (compared to -1.66 ‰) and CIEs of -1.01‰ (-1.27 ±0.29 ‰, "mean-mean") or -1.37‰ (-1.74 ±0.46 ‰, "mean-most negative value").

[Figure]

[Figure]

3. Something is not quite right about the PETM thickness. There are 180 meters for the Lower Eocene in the section (fig 2, and fig 3 of Brandes et al paper). Early Eocene spans 8.2 my, that would give 45Ky per meter, and 10 meters would be about half a million years, that is twice to three times the span of the PETM. Brandes did a basin modeling analysis and ages for the modeling were derived from "interpolated from literature data" (Table 2). And the "literature data: in the paper refers to "According to these sources, a sample from slightly above the Main Seam has been assigned to dinoflagellate zone D5b (Ahrendt et al., 1995), the base of which is dated to about 54.8 my b.p. (Kothe, 2003). The latter age may thus serve as an approximation for the base of the Main Seam (Gurs et al. ,2002) and the

Schoningen Formation. A radiometric age of 46 my b.p. and dinoflagellates indicative of zone D9na have been derived from slightly below the Heidberg unit by Ahrendt et al. (1995) and Lietzow Ritzkowski (1996, 2005a), but a discrepancy is caused by the fact that according to Kothe (2003), dinoflagellate zone D9a ranges from 50.51 to 48.5 my b.p".

It seems that 1) the Brandes modeling is not independent as it relies on the ages, and 2) ages themselves have discrepancies. If zone D5b is slightly above Main seam, it would imply for the PETM to be below, not above the Main Seam. The most difficult task when working the PETM is having strong evidence of the precise stratigraphic location of the PETM. It should be fully demonstrated. This manuscript requires a much stronger argument to support the position of the PETM. For instance, it would very useful to have the D13C1 record of the entire early Eocene to see if the long-term negative excursion of the EETM is recorded.

- Both points raised in this comment are correct: 1.) the modeling presented by Brandes is not independent of the age constraints and 2.) the ages have its own discrepancies.
- Concerning 1.) Brandes et al. (2012) performed basin modelling of the Schöningen basin/rim syncline with a focus on the basin burial history. They found: "*A clear decrease in sedimentation rates through time can be observed. In the early phase of basin evolution, the sedimentation rates were high, with values of 60–80 m/Myr during the formation of the Main Seam and Seams 1 and 2. Subsequently, the sedimentation rates decreased to 32–56 m/Myr during the formation of Seams 3–9.*" (Brandes et al. 2012, Basin Research, 24, p. 709). Therefore we applied a sedimentation rate of 60-80 m/Ma to our records (comprising Seam 1 and Seam 2), rather than an average sedimentation rate.
- However, due to major rewriting of the manuscript now focusing on the description of the CIE rather than its assignment to the PETM, we do not feel that calculation the event duration in such an environment is robust or required and, therefore, we deleted parts of this paragraph.
- Concerning 2.). The weakness of the age constraints has also been noted by the reviewers and in short comment #2. We therefore address this more prominently in the manuscript by including a separate section (section 2, p. 3/4).

4. A simple inspection of the pollen data is not enough. The palynological analysis requires a statistical test to support the conclusion that there is little change across the PETM. For example, a multidimensional scaling could be run using the Chao-dissimilarity index to test if the differences within PETM samples are as high as differences among pre, post and PETM samples.

- We appreciate this valid request and followed the suggestion of the reviewer. Non-metric multidimensional scaling was performed for the pollen and spore data set without algae using the Bray-Curtis dissimilarity and the Wisconsin double standardized raw data values.
- More details are given in the method section (section 3.3: p. 5, ln. 30 - p. 6, ln. 9) and the results section (section 4.3, p. 11, ln. 1-9 and p.12, ln. 15-18). In addition to this, we included a new figure 6 showing the results of the NMDS analysis.

Page 2 line 9: A. augustum

- We followed the suggestion of the reviewer and inserted "augustum" (p. 3, ln. 23).

Page 5: line30 to Page 6 line 16: this part should be rewritten with more care, in the view of the main comment about the definition of this interval as the PETM

- We appreciate this valid request. As stated above, we rewrote the manuscript in order to 1) describe the detected CIE, 2) give hypothesis about the interpretation of the CIE and 3) make a more tentative comparison of the existing lignite records (see reply above).

Page 6 line 12: add a reference here for the PETM duration

- Due to the major changes in this section of the manuscript, the questionable paragraph has been deleted.

---

## Author Comment (AC3) · 30 Jun 2019

**Review 1**
**Anonymous Referee #1**

The manuscript by Methner et al. present new interesting datasets that represent a great contribution to the available terrestrial records from Eocene sections. The authors present new d13C_TOC and TOC (%) data, as well as palynological data from the mid-latitude coastal site of Schoningen. The main conclusion is the identification of the PETM in the CIE represented by the onset of a negative excursion in d13C TOC in Seam 1 and a positive shift in Seam 2, as well as the presence of Apectodium in the marine interbed comprised between the two lignite seams.

The paper is well-written and properly structured and addresses interesting scientific questions which fit the scope of publications in Climate of the Past. Figures and tables are of good quality and relevant to the manuscript. However, I do recommend to rethink and reframe the manuscript because of the following problems.

1) The identification of the PETM. My main concern is that the evidence brought forward to define the CIE as the PETM should be discussed with care. The authors briefly acknowledge that the identification of the PETM in this interval has been suggested before by Riegel et al., (2012) but fail to discuss the possible pitfalls of this assumption:

a) Ages: possible age discrepancies in the dating of the section should be discussed in the manuscript. I refer to the detailed comment by Carlos Jaramillo, who has already noted that "independent" dating by Brandes et al (2012) is relying on ages related to the dinoflagellate zones. This should be addressed in a section of the manuscript

- This is a valid and important request and in concert with the concerns of review #2 and the two posted comments stating that the age constraints for the Schöningen record is rather weak and suggesting the possibility that the detected CIE reflects another Early Eocene hyperthermal event, e.g. the ETM2, rather than the PETM.
- Given that this is a major concern of all reviews/comments, we now address this problem more upfront and therefore, included a new section (section 2; p. 3, ln. 13 – p. 4, ln. 10) to give detailed information about the available age constraints and difficulties in local correlations (between the Schöningen open pit and the Emmerstedt drill core).

b) Thickness: a 10-m thick CIE; this requires a very high sedimentation rate (as noted in line 13) but is this reasonable? How is this changing from one lithology to the other in the transition from lignite to the marine interseam? Also, if we accept an almost linear sedimentation rate (_0.5 Myr/m) for the whole section, then the duration of the CIE doubles. How can you exclude that this is not the case?

- For the sedimentation rates we refer to the basin modeling study of Brandes et al. (2012), who detected a high mean sedimentation rate of 60-80 m/Ma. (see also reply to comment of Carlos Jaramillo, issue 3). We cannot exclude that the sedimentation rates differ from the reported ones, but currently cannot refine these as this would require further sedimentological and/or geochronological investigations that go beyond the scope of this study. However, Brandes' basin model relies on the same

age constraints as our study, the argument that the duration of the CIE agrees with the duration of the PETM is circular.

- Similar to the reply to comment#1 and #2: Due to major rewriting of the manuscript now focusing on the description of the CIE rather than its assignment to the PETM, we do not feel that calculation the event duration in such an environment is robust or required and, therefore, we deleted parts of this paragraph.

And how can you exclude that the CIE is not, for example, the expression of ETM2/H2 hyperthermals (which would together last about 400 kyr)?

- Currently, we cannot exclude this possibility that the CIE reflect an Early Eocene hyperthermal (ETM2 or any other) (see discussion about the weakness of the available age constraints). Please see also the reply below (general suggestions) and the reply to comments by Gerald Dickens.

c) Hyperthermals: the acme of the Apectodium is the strongest evidence use to correlated this interval with the PETM. This is quite a compelling evidence, however, as noted by Jerry Dickens in his comment, Apectodinium augustum is the diagnostic species found in PETM section. Is there evidence for its presence?

- Unfortunately, there is no evidence for Apectodinium augustum (now assigned to Axiodinium; see text) in the marine Interbed 2, but we discuss this and also the possible reasoning for the lack of the species in the newly added section 2. The lack of A. augustum persists in the over- and underlying Interbeds. This may, therefore, arise from the paleoecological setting (see discussion in section 2). At the moment we cannot resolve this question and, thus, address this issue more prominently (new section 2) and in an open way to the reader (p. 7, ln. 20-24; p. 7, ln. 31 – p.8, ln. 6; p. 8, ln. 16-20; p. 12, ln. 25-29).

In general, I recommend incorporating those points in the discussion, discussing more in detail all the possible pitfalls associated with a univocal identification of the PETM as well as the strong evidence in support of it.

- We appreciate this suggestion very much. As stated above and in the replies to the two comments, we now address the problematic age assignment more prominently (own section 2).
- To provide more clarity to the reader, we first describe the CIE in a more general way, not relating it to a particular hyperthermal and then discuss the possibility that the detected CIE reflect another Eocene hyperthermal (not the PETM) (section 4.1, p. 7, ln. 20-24).
- Especially the comparison to the near-by paleo-North Sea lignite records (Cobham and Vasterival) has interesting implication, in case that the detected CIE is not related to the PETM. The similarities between the records may arise from the fact that they represent the same hyperthermal (PETM or any other Early Eocene hyperthermals) or that different hyperthermals have similar effects to mid-latitudinal wetlands in the paleo-North Sea realm. We now clearly state that there is no unequivocal proof that the three lignite records reflect the same event (section 4.2: p. 8, ln. 16-20 and p. 7, ln. 31 – p.8, ln. 6 and in section 4.4: p. 12, ln. 25-29).

I also suggest following Carlos Jaramillo's comments on the raw palynological counts and the TOC vs d13C TOC correlation plot.

- Concerning the raw palynological counts, we appreciate the valid request and now provide the raw palynological counts in the Appendix (section SI3, table SI3).
- Concerning the %TOC vs $\delta^{13}C_{TOC}$ correlation plot, we gave a detailed response in the comment to Carlos Jaramillo (see copy below):

- We appreciate this comment as this method is a valid approach for carbon isotope ratios of soil TOC and %TOC (Wing et al., 2005) and very interesting to apply this to our dataset.
- In brief, we find that a **direct assignment of this approach is not possible due to the very different nature of the two settings** (lignite seams/marine sediments and mud-rock paleosols). We deduced the %TOC vs. $\delta^{13}C_{TOC}$ relationships in our data set and found very different results to Wing et al. (2005). In particular, we found no relationship across the whole data set ($R^2$ = 0.002), a moderate positive linear relationship for samples with %TOC >50% (lignite seams) ($R^2$ = 0.208), and again no relationship for samples with %TOC <50% (marine interbed) ($R^2$ = 0.039). However, if applying these "transfer functions" (for details see below), we **maintain the deduced negative carbon isotope excursion**.
- In summary, we feel that exploring on such a relationship in lignite samples and/or modern peat samples would go beyond the scope of this paper, but consider this an interesting approach that needs further investigation.

   In detail:
- Wing et al. (2005) found a strong negative exponential relationship between %TOC (independent variable) and $\delta^{13}C_{TOC}$ in mud-rock paleosol samples in the sub-groups of PETM ($R^2$ = 0.623) and Paleocene-Eocene ($R^2$ = 0.618) samples. For comparison, modern soil samples show a coefficient of determination of 0.952.
- Plotting our data (with the %TOC as the independent variable and $\delta^{13}C_{org}$ as the dependent variable), **no such relationship could be determined** (see plot below). The coefficient of determination $R^2$ is 0.002 and a F-test revealed no statistical relationship between these two variables.
- Only considering lignite values with %TOC > 50% yields a moderate positive linear relationship ($R^2$ = 0.208) that is statistically significant. Considering values of %TOC < 50% (marine interbed, typical %TOC <10%), we found a weak negative linear relationship ($R^2$ = 0.039) that is statistically not significant (F-test).
- Applying the deduced regressions (as transfer functions) -*even though the second regression is not significant*- to our data set in order to plot the residuals from expected $\delta^{13}C_{TOC}$ values, **maintains the deduced negative carbon isotope excursion**. The magnitude is slightly reduced at the CIE onset with -1.22 ‰ (compared to -1.66‰) and CIEs of -1.01‰ (-1.27 ±0.29 ‰, "mean-mean") or -1.37‰ (-1.74 ±0.46 ‰, "mean-most negative value").

[Figure]

[Figure]

Minor comments:

Page 1 Lines 6-9: The release of isotopically light carbon was most likely a feedback of the long-term warming rather than the cause.

- Agreed. We acknowledge that this phrasing is misleading and has been changed accordingly. In addition to this, the major revisions of the manuscript required major rewriting of the abstract.

Rephrase. Line 21-23: another problem associated with the interpretation of sources and mechanisms is local signals in different records

- We followed the suggestion of the reviewer and rephrased this sentence (see also the reply to a similar comment by Gerald Dickens).

Page 2 line 9: A. augustum

- We followed the suggestion of the reviewer and inserted "augustum" (p. 3, ln. 23).

Page 5: line30 to Page 6 line 16: this part should be rewritten with more care, in the view of the main comment about the definition of this interval as the PETM

- We appreciate this valid request. As stated above, we rewrote the manuscript in order to 1) describe the detected CIE, 2) give hypothesis about the interpretation of the CIE and 3) make a more tentative comparison of the existing lignite records (see reply above).

Page 6 line 12: add a reference here for the PETM duration

- Due to the major changes in this section of the manuscript, the questionable paragraph has been deleted.

---

## Author Comment (AC4) · 30 Jun 2019

**Review 2**

**Joost Frieling (Referee),** j.frieling1@uu.nl

Review of Methner et al. Clim. Past. – first version.

**Summary**

The manuscript presented provides (1) new stable carbon isotope data across part of the Schoningen lignites (2) detailed palynological assemblage data from the same interval and (3) a comparison with other NW European lignites which are argued to be time-equivalent. With these data, the authors aim to resolve the response of wetland/peat systems to global warming across the Paleocene-Eocene Thermal Maximum.

I agree with the authors that it is of great importance to understand the behavior of wetland/peat systems in warm(ing) climates and this is an appropriate subject for this journal.

The results lead the authors to several general conclusions about the analyzed section, most of which I have no serious concerns about, including how the influences of increased fire activity (seasonal drying) and drowning due to higher relative sea level led to the (local) demise of these peat mire systems.

However, I think it is uncertain whether the comparison to other records is solid and this potentially has major implications for the regional picture and the extrapolations to past and present global warming. The regional comparison is mainly built on the assumption that the sections are time equivalent (major concern #1) and that relatively small variability (~2‰) in carbon isotopes in bulk organic matter within heterogenous lithological columns is indicative of the PETM or a similar hyperthermal event (major concern #2).

- This is a major issue that has been similarly addressed by the comments and reviews#1. We acknowledge that in the first version of the manuscript a clear assignment of our detected CIE to the PETM might have been too bold.
- Identical reply to comment by Gerald Dickens: We acknowledge this by
  - (1) adding a whole new section (now section 2) to the manuscript in order to describe the available age constraints and the pitfalls/discrepancies of them in more detail (section 2; p. 3, ln. 13 – p. 4, ln. 10)
  - (2) being more careful in our wording while describing solely the CIE and not unequivocally relating the CIE to the PETM, ETM2 or any other Early Eocene hyperthermal (section 4.1; p. 7, ln. 20-24).
  - (3) discussing tentatively the possible assignment of the CIE to the PETM vs. any other Early Eocene hyperthermal (section 4.2; p. 8, ln. 16-20)
- We rephrased major parts of section 3.1, now section 4.1 (p. 6/7) and included a statement that we compare the European wetland records despite the possibility that they may reflect different hyperthermal events in section 4.2 (p. 7, ln. 31 – p.8, ln. 6) and 4.4 (p. 12, ln. 25-29).
- In order to place our results in a more regional framework, we still perform the comparison with nearby lignite sites (Cobham, Vasterival) in which the reported CIEs have been assigned to the PETM. We feel, in agreement with this comment, that this comparison might still be valuable to detect similar behaviors of these Paleogene wetlands during carbon cycle perturbations. However, we now clearly state that there

is no sufficient proof that these records are time-equivalent as they all have their limitations when it comes to age assignment (in section 4.1, 4.2 and 4.4).

- In addition to this, the comparison to the near-by paleo-North Sea lignite records (Cobham and Vasterival) has interesting implications, in case that the detected CIE is not related to the PETM. The similarities between the records may arise from the fact that they represent the same hyperthermal (PETM or any other Early Eocene hyperthermals) or that different hyperthermals have similar effects to mid-latitudinal wetlands in the paleo-North Sea realm. We now address this in section 4.4 (similar to reply to review#1). However, we only state observations at this point and we try to be as careful as possible without judging upon the pitfalls of other studies.
- A minor remark: The whole lithological column is indeed heterogeneous (alternating lignite seams and marine interbeds) however, the discussed CIE occurs only within the lignite (Seam 1) and independent of any detectable lithological change. According to Collinson et al (2003), the same applies to the Cobham section.

**Major concern #1**

Dating lignite sequences is notoriously difficult and extreme caution is warranted when correlating these deposits to geologically very short, in this case 50-200 kyr, events. In an earlier publication (Riegel et al. 2012), three authors of this manuscript show that another part of the same sequence (seam #6) is associated with substantial amounts of *Apectodinium*. Also below the here presented *Apectodinium* acme, there is a smaller distinct abundance spike in *Apectodinium* (bottom marine interbed #1). This spike, based on previous correlations that the authors also mention here (p. 3, lines 12-16), could be placed close to the P/E boundary, but also still **above** the P/E.

Importantly, high percentages of *Apectodinium* in other Paleocene and Eocene successions from mid and high latitude settings are not strictly limited to the PETM or even hyperthermal events (examples include Bijl et al., 2013; Frieling et al., 2018; Heilmann-Clausen, 2018; Sluijs et al., 2011). I think the authors claim in p. 3, lines 8-12 should be rephrased to accommodate these observations and potential implications thereof.

- This is a valid request and has similarly addressed by the other reviewer and in the short comments. In order to address the problematic age constraints and to discuss the pitfalls of using the Apectodinium acme to infer the PETM, we included a new section (section 2; p. 3, ln. 13 – p. 4, ln. 10).

Assuming the CIE is not an artifact of preservation or source changes (see also below) there is as much evidence to connect the carbon isotope excursion here to the PETM as to any other early Eocene hyperthermal. If the age of the analyzed section cannot be constrained sufficiently, a detailed comparison with other lignites (Vasterival / Cobham) including the carbon isotope changes would become more complicated and would require more nuanced statements.

- We fully agree with this statement. We, hopefully, fulfilled this requirement and provide more clarity to the reader by (1) describing the CIE in a more general way (section 4.1), (2) discuss the possible assignment to the PETM vs. any other Early Eocene hyperthermal (section 4.2), and (3) make the comparison to the other lignite

records pointing to the striking similarities, but with a more tentative interpretation of these (section 4.2 and 4.4) (see reply to review#1 and short comment #2).

Hopefully, the authors can show the presence of PETM marker species, either dinocysts (*Apectodinium augustum*, high variability of morphology within the *Apectodinium*/Wetzellioid group (e.g. Iakovleva, 2016) or pollen (comparison with Eldrett et al., 2014; Willumsen, 2004). Likewise, if there are identifiable ash layers within the sequence this could be a welcome addition to resolve the local stratigraphy (e.g. (Heilmann-Clausen, 2018; Jones et al., 2019; Westerhold et al., 2009). If the correlations to other localities cannot be made with confidence, I think the comparison with other lignites and pollen studies should be rewrittento paint a much more general picture (see also point 2).

- Firstly, as already stated above, we now address the available age constraints more prominently in the new section 2 (p. 3, ln. 13 – p. 4, ln. 10).
- Secondly, there is no ultimate proof that our detected CIE is correlative to the CIEs in the lignite records (that have been assigned to the PETM). However, we feel confident that a comparison still yields interesting results (see reply above).

**Major concern #2**

The carbon isotope signal is from integrated bulk organic matter, implying that large changes can occur if any of the following factors play a role.

1. There is likely to be a difference in the marine/terrestrial fraction within the bulk organic matter across the lithological transitions, but this may not be entirely limited to these transitions. Hopefully, the authors can show from their palynological assessment how the marine/ terrestrial fraction varies across the lithological transitions. This is of vital importance as marine and terrestrial organic matter sources can be offset by ~4‰ (cf. (Sluijs and Dickens, 2012)). The palynological data already allows a preliminary assessment of marine/terrestrial fractions, possibly without any further analyses.

- This is an important point when assessing bulk organic carbon isotope values. There are several lines of evidence to exclude mixing between marine and terrestrial derived organic matter as the main reason for the CIE:
    - The palynological data records rather defined transitions between the terrestrial peat deposits and the marine clastic interbeds.
    - The transitions between the lignite seams and the marine interbed are rather abrupt within a cm-dm scale.
    - Samples that experienced possible mixing (base of Interbed 2) have been discussed in the manuscript (p. 5, ln. 26-29, now p. 7, ln. 6-7). Most obvious is the %TOC data with %TOC > 50% typical for the lignite seams.
    - The deduced CIE occurs fully within the lignite Seam 1 not showing any indication of mixing at this stratigraphic level.
- As this is a potential major pitfall for the interpretations, we discuss this issue in more detail in the manuscript (p. 6, ln. 29 – p. 7, ln. 5 and p. 7, ln. 15-19)

2. The preservation-regime may be very different in marine and terrestrial environments, which could also skew the relative fractions of marine (aquatic) and terrestrial OM, the latter being more resistant to oxidation (Huguet et al., 2008).

- Similarly to the arguments above, the discussed onset of the CIE occurs within the lignite seam, unrelated to the marine interbed.
- The $\delta^{13}C_{TOC}$ values during the inferred CIE (upper part of Seam 1) and the marine interbed are indeed different. We attributed this to a possible mixing, however, following the comment of the reviewer (see below), we reevaluated this statement and now elaborate on the possible different fractions of the OM (p. 6, ln. 29 – p. 7, ln. 5).

3. The authors mention potential bacterial influence on carbon isotope signals across the "CIE" events and show a comparison with other sources, showing lignites are essentially recording a muted isotope signal. This also ties in with point #1 and should be assessed in more detail. I encourage the authors to explore alternative possibilities of forming a CIE in a lignite-marine intercalated sequence.

- As mentioned above, we cannot detect any marine influence in our CIE. However, to improve/clarify the information to the reader, we now included an additional paragraph in the main text (p. 6, ln. 29 – p. 7, ln. 5 and p. 7, ln. 15-19).

4. The completeness of the section is not addressed in detail at the moment and should be expanded upon. In laterally heterogenous sequences, which include sharp lithological transitions, it seems likely that there are smaller and/or larger hiatuses, which appear in the record as a sharp isotope shift, if imposed on a long-term isotope trend.

- This is a valid request. As mentioned above, the transitions between the seams and the marine interbeds are abrupt, however, there is no evidence for any major hiatus. But again, the onset of the CIE occurs within the lignite seam and we could not detect any marine influence at this stratigraphic level. We now address this issue more upfront in the method section (p. 4, ln. 20-22) and again in section 4.1, p. 7, ln. 15-19.

5. The carbon isotope signature of charcoal can be depleted by up to 2‰ relative to the source material (Ascough et al., 2008). As such, even without source or vegetation changes, a change in fire regime could result in a negative CIE in a lignite record. This is particularly worrying if the CIE onset coincides with a charcoal spike in the record and raises the question whether such smaller carbon isotope trends in these deposits are perhaps always locally induced.

- This is an interesting remark that we appreciate very much. Indeed, an increased amount of charcoal has been detected in Seam 1 (and Seam 2) compared to the subsequent seams (Robson et al. 2009).
- However, this would mean that almost the entire CIE (89% given the CIE onset of -1.77‰ and 63% given a CIE of -1.26‰ by the means) would be due to the presence of charcoal in the uppermost samples of Seam 1. Such a change in the charcoal content in the upper samples of Seam 1 has not been observed. Instead, a similar amount of charcoal has been recorded in all of these samples (see Appendix SI1).

This also applies to most other previously analyzed lignite / marine sandstone records that have been interpreted in similar manners, but without scrutiny of the isotope trends. I think with the current knowledge, the authors can contribute significantly to a much more solid discussion on interpreting carbon isotope records in lignite sequences.

- We thank the reviewer for the detailed comments on the stable isotope data. We included new paragraphs in section 3.1 (sampling) and 4.1 (CIE discussion) to address the issues raised above.

**Minor comments**

P2. Lines 6-9. The global warming is around 4-5 oC, see (Dunkley Jones et al., 2013; Frieling et al., 2017). Local warming is occasionally amplified to ~10oC (e.g. Schoon et al., 2015).

- We followed the suggestion of the reviewer and rephrased this together with a major part of the introduction to account for the shifted foci of the manuscript (p. 1, ln. 8-10).

P2. Line 11. "two-step" is confusing here, can be removed.

- We followed the suggestion of the reviewer and changed the wording accordingly (p. 2, ln. 11).

P2. Lines 23-24. The transition between these paragraphs is rather abrupt and the two seem somewhat disconnected. Can you clarify the reasoning here?

- We appreciate this comment very much. Due to major restructuring of the manuscript (see also above), this issue has hopefully been resolved (c.f. Introduction, p. 2/3).

P2. Lines 25-26. On what time scales are these wetlands important for carbon cycling?

- This is not a trivial answer. Wetlands act on very different timescales either as sources or as sinks of carbon. This depends on the climatic regime, the organic input (down to species level, e.g.), hydrological regime (groundwater influences or fluvial export of DIC), as well as the depositional environment. In general, wetlands have high carbon turnover rates (tens of days). Whereas, burning of peat releases vast amounts of carbon to the atmosphere within days, weeks or even month. Carbon storage acts on longer timescales by building up peat and burial of peat to form lignite (and coal + natural gas), e.g. Eocene or Miocene lignite deposits.
- Given that wetlands can act as both sources and sinks/storage of carbon (dep. on timescale and climatic as well as depositional regimes, we feel that including such discussions into the manuscript would be beyond the scope of the paper.

P3. Lines 12-21. I have some difficulty following the reasoning here: at first, you state the correlations placed the PETM within or below the main seam but then quote an age (54.8-54.4 Ma) which does not align with that statement or the analyses of a lignite/marine interbed above the interval that was originally correlated to the PETM?

- As stated above, we address this in a new section (section 2, p. 3, ln. 13 – p. 4, ln. 10).

P4. Lines 28-29. Palynological treatment with hydrogen peroxide and KOH will result in loss of some fragile palynomorphs, including dinoflagellate cyst species. If present, Peridinioids with hexa-2a archaeopyles (e.g. *Senegalinium*, *Phthanoperidinium*, *Lejeunecysta* etc.) are probably affected worst (e.g. Zonneveld et al., 2019). Unfortunately, these are also low-salinity tolerant species. While it is difficult to assess what the exact influence of this is on the assemblage study, it should at least be acknowledged that there may be an effect.

- We think that this is a justified remark of the reviewer and addressed this in the method section (p. 5, ln. 20-21). In several previous projects we have received rich and diverse dinocyst assemblages by the same procedures without any sign for selective degradation (Lenz 2005; Lenz et al 2007; Riegel et al. 2015).

P7. Lines 28-30. The exact opposite should be the case for the bulk organic matter, given that Paleogene marine organic matter is more 13C-depleted (see Sluijs & Dickens, 2012).

- We appreciate this comment and elaborate on this in the manuscript text (p. 6, ln. 29 – p. 7, ln. 5) as we feel that this observation even strengthen the description of an early Paleogene CIE.
- Our $\delta^{13}C_{TOC}$ values from the marine interbed hit exactly the given "background marine endmember" $\delta^{13}C$ value (Sluijs and Dickens, 2012), which may call for a fully marine but non-PETM signal. Marine PETM-related $\delta^{13}C_{TOC}$ values (at least at the Lomonosov Ridge) are ~3 ‰ lower if they are largely unaffected by terrestrial carbon input, but tend to be almost identical to the "background marine endmember" value under high terrestrial TOC input (~ -26 to -27 ‰).
- Given the near coastal setting of the Schöningen locality during the early Paleogene we can expect a significant terrestrial contribution to the TOC. Indeed can be observed by the presence of terrestrial palynomorphs in the marine interbed (Fig. 5). The average $\delta^{13}C_{TOC}$ value of -27.7 ‰ of interbed 2 falls exactly into the range of values for PETM-related $\delta^{13}C_{TOC}$ values with high (~80%) terrestrial contribution. Thus, the "elevated" (compared to the lignite $\delta^{13}C_{TOC}$ values) $\delta^{13}C_{TOC}$ values of the clastic interbed are in good agreement with the CIE/PETM-related marine $\delta^{13}C_{TOC}$ values of Sluijs and Dickens (2012) and thus support our finding that the CIE at Schöningen comprises the whole interbed 2 and extended into seam 2.

P9. Line 22-30. How similar are these assemblages to other localities in the same area across the PETM (e.g. Eldrett et al. 2014, Willumsen, 2004)?

- Both papers are now considered in the main text (p. 10, ln. 22 and p. 12, ln. 14). We report now that in some Late Paleocene/Early Eocene records around the North Sea basin significant influences of short-term thermal events such as the PETM on the composition of the vegetation have been recognized (e.g. Eldrett et al., 2013). However, for the Schöningen record, respectively, for the relatively short section that we have analyzed so far we have indications that the vegetation changes only follow natural successions and are not related to other factors such as climate influences. Therefore, we mention that we need the long-term record of the Schöningen succession to identify the climate influence (p. 12, ln. 20-23). This is part of our ongoing study of the Schöningen record.
- The palynomorph assemblages from Schöningen reveal close similarities with the Danish flora reported by Willumsen (2013) (p. 10 ln. 21-22). However, this is only briefly touched upon as the main focus of the paper lies in the comparison of the lignite records/wetland deposits.

P10. Lines 17-20. Can it be excluded that every signal here is autocyclic, and simply reflects the natural progression of a wetland system?

- We do not intent to exclude this. Instead we clearly state in the manuscript we state that this follows a natural transition (p. 1, ln. 21-22; p. 12, ln. 22-23; p. 13, ln. 26-27).

Figures 2, 3 & 5: It would be helpful for the reader to have a detailed correlation between the section analyzed for carbon isotopes and palynology and/or the carbon isotope profile should be included in Figure 5. At present, the different height/depth scales of the two analyzed sections make it difficult to see what is connected to what.

- We thank the reviewer for this valid remark and the opportunity to improve the figures and provide better understanding for the reader. We changed the figure 5 accordingly.
- In particular, Figures 2 and 3 show the detailed carbon isotope data with the identical stratigraphy. Thus, we did not change these figures. However, we follow the suggestion of the Reviewer and included the isotopic data. The sections have been correlated according to the top of Seam 1 and base of Seam 2.

**References**

Eldrett, J.S., Greenwood, D.R., Polling, M., Brinkhuis, H., Sluijs, A., 2014. A seasonality trigger for carbon injection at the Paleocene–Eocene Thermal Maximum. Clim. Past 10, 759–769.

Lenz, O.K., 2005. Palynologie und Paläoökologie eines Küstenmoores aus dem Mittleren Eozän Mitteleuropas- Die Wulfersdorfer Flözgruppe aus dem Tagebau Helmstedt, Niedersachsen. Palaeontographica Abteilung B 271, 1-157.

Lenz, O.K., Wilde, V., Riegel, W., 2007. Recolonization of a Middle Eocene volcanic site: quantitative palynology of the initial phase of the maar lake of Messel (Germany). Rev Palaeobot Palyno 145, 217-242.

Riegel, W., Lenz, O.K., Wilde, V., 2015. From open estuary to meandering river in a greenhouse world: an ecological case study from the middle Eocene of Helmstedt, northern Germany. Palaios 30, 304-326.

Robson, B.E., Collinson, M.E., Riegel, W., Wilde, V., Scott, A.C., Pancost, R.D., 2015. Early Paleogene wildfires in peat-forming environments at Schöningen, Germany. Palaeogeography, Palaeoclimatology, Palaeoecology 437, 53-62.

Sluijs, A., Dickens, G.R., 2012. Assessing offsets between the $\delta^{13}C$ of sedimentary components and the global exogenic carbon pool across early Paleogene carbon cycle perturbations. Global Biogeochem Cy 26.

Willumsen, P.S., 2004. Palynology of the Lower Eocene deposits of northwest Jutland, Denmark. Bull. Geol. Soc. Denmark 51, 141–157.

---

## Author Comment (AC6) · 30 Jun 2019

Plaese see the anticipated changes in the revised manuscript.

Please also note the supplement to this comment:
https://www.clim-past-discuss.net/cp-2019-20/cp-2019-20-AC6-supplement.pdf